# ANO: A Unified RL Framework for Robust Policy Optimization

## Abstract

Proximal Policy Optimization (PPO) dominates deep RL but faces a fundamental dilemma. Its "hard clipping" mechanism discards valuable gradient information from outliers, leading to sample inefficiency. Conversely, removing clipping (as in SPO) exposes optimization to unbounded gradients, causing significant instability and hyperparameter sensitivity. To resolve this, we establish a Unified Trust Region Framework that generalizes existing objectives. Within this framework, we derive Anchored Neighborhood Optimization (ANO) based on a set of design principles. We identify that the failure of standard policy gradients stems from a misapplication of gradient influence on outliers. We propose the Redescending Influence Principle, a paradigm shift from monotonic penalties (SPO) and hard-thresholding (PPO) to dynamic outlier suppression, and prove its necessity for stability in high-variance stochastic optimization. Theoretically, we prove ANO possesses the minimal structural complexity required for robust optimization. Empirically, ANO achieves state-of-the-art performance on MuJoCo benchmarks, significantly outperforming PPO and SPO. Notably, ANO demonstrates superior stability, preventing policy collapse even under aggressive hyperparameters (e.g., learning rates $3\times$ larger than standard) where PPO fails completely[1].

## 1. Introduction

Deep Reinforcement Learning (DRL) has achieved remarkable success in areas such as gaming, robotic control, and language model alignment (Mnih et al., 2015; Silver et al., 2016; 2018; Vinyals et al., 2019; Ye et al., 2020; Makoviychuk et al., 2021; Rudin et al., 2022; Heess et al., 2017;

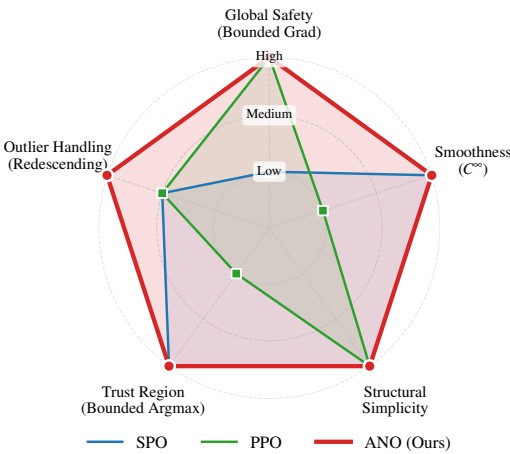

*Figure 1.* **Comparison of shaping function properties.** We evaluate PPO, SPO, and ANO across five dimensions. While all methods are structurally simple, PPO and SPO suffer from topological degeneracy (0 inflection points), leading to either information loss or instability. **ANO (Ours)** introduces the **minimal necessary topological structure** (one inflection point) to balance safety and smoothness without superfluous complexity.

Schulman et al., 2015b; Ouyang et al., 2022; Black et al., 2023). However, Policy Gradient (PG) methods face a fundamental trade-off between update stability and learning efficiency. Trust Region Policy Optimization (TRPO) enforces stability through second-order constraints but is computationally expensive (Schulman et al., 2015a). Proximal Policy Optimization (PPO) addressed this by introducing a first-order clipping mechanism, achieving wide adoption due to its simplicity (Schulman et al., 2017).

Despite its popularity, PPO's theoretical justification diverges from practice. Empirical audits reveal that "hard clipping" frequently fails to enforce the trust region (Ilyas et al., 2018; Wang et al., 2020) and relies heavily on code-level optimizations (Engstrom et al., 2020). While recent methods like SPO (Xie et al., 2025) propose more strong constraints but leading to unbounded gradients and other problems.

To resolve this, we propose the **Unified Trust Region Framework**, identifying that an ideal estimator must be *anchored* near the trust region for efficiency but possess *redescending* gradients for outliers to ensure safety. Based on this, we introduce **Anchored Neighborhood Optimization (ANO)**. As shown in Figure 1, unlike PPO's "flat tail"

[1]Anonymous Institution, Anonymous City, Anonymous Region, Anonymous Country. Correspondence to: Anonymous Author <anon.email@domain.com>.

Preliminary work. Under review by the International Conference on Machine Learning (ICML). Do not distribute.

[1]Code link: https://anonymous.4open.science/r/ano-F818

(unbounded maximizers) or SPO's "unbounded growth", ANO employs the **minimal sufficient topology**, a single convexity change, to strictly enforce the trust region without explicit clipping.

We summarize our main contribution as follows:

- **Unified Framework:** We establish a unified perspective that connects trust region methods with gradient shaping functions and propose 5 design principles for ideal shaping functions.
- **Anchored Neighborhood Optimization (ANO):** We derive ANO, an algorithm that strictly adheres to these principles. We provide rigorous proofs showing that ANO possesses the minimal topological complexity required (a single convexity change) to satisfy robustness constraints.
- **SOTA Performance & Robustness:** ANO not only outperforms PPO and SPO in various reinforcement learning tasks, but also exhibits exceptional stability. Crucially, ANO prevents policy collapse even under aggressive learning rates ($1 \times 10^{-3}$), where PPO fails catastrophically.

## 2. Preliminaries

### 2.1. Reinforcement Learning Framework

We formulate the sequential decision-making problem using the standard Markov Decision Process (MDP), denoted as a tuple $\mathcal{M} = (\mathcal{S}, \mathcal{A}, r, \mathcal{P}, \rho_0, \gamma)$. Here, $\mathcal{S}$ and $\mathcal{A}$ denote the state and action spaces, respectively. The transition dynamics are governed by $\mathcal{P} : \mathcal{S} \times \mathcal{A} \times \mathcal{S} \mapsto [0, 1]$, and the reward function is defined as $r : \mathcal{S} \times \mathcal{A} \mapsto \mathbb{R}$. $\rho_0$ represents the initial state distribution, and $\gamma \in (0, 1)$ serves as the discount factor.

In this framework, an agent interacts with the environment according to a stochastic policy $\pi$, generating a trajectory $\tau = (s_0, a_0, r_0, \dots)$ where $a_t \sim \pi(\cdot|s_t)$ and $r_t = r(s_t, a_t)$. The primary objective is to maximize the expected discounted cumulative return:

$$\eta(\pi) = \mathbb{E}_{\tau \sim \pi} \left[ \sum_{t=0}^{\infty} \gamma^t r_t \right], \tag{1}$$

where $\mathbb{E}_{\tau \sim \pi}$ denotes the expectation over trajectories induced by the policy $\pi$ (i.e., $s_0 \sim \rho_0(\cdot), s_{t+1} \sim \mathcal{P}(\cdot|s_t, a_t)$). We define the standard action-value function $Q_\pi$ and state-value function $V_\pi$ as:

$$Q_\pi(s_t, a_t) = \mathbb{E}_{s_{t+1}, a_{t+1}, \dots} \left[ \sum_{l=0}^{\infty} \gamma^l r(s_{t+l}) \right], \tag{2}$$

$$V_\pi(s_t) = \mathbb{E}_{a_t, s_{t+1}, \dots} \left[ \sum_{l=0}^{\infty} \gamma^l r(s_{t+l}) \right]. \tag{3}$$

Accordingly, the advantage function is given by $A_\pi(s_t, a_t) = Q_\pi(s_t, a_t) - V_\pi(s_t)$.

### 2.2. Policy Improvement Lower Bound

Standard policy gradient methods often suffer from sample inefficiency and hyperparameter sensitivity. To mitigate these issues, Trust Region Policy Optimization (TRPO) (Schulman et al., 2015a) establishes a theoretical foundation for monotonic improvement. This relies on the Performance Difference Theorem originally introduced by Kakade & Langford (2002):

**Theorem 2.1.** *(Kakade & Langford, 2002) Let $\rho_\pi(s)$ be the normalized discounted visitation distribution induced by policy $\pi$. For any two policies $\pi$ and $\tilde{\pi}$, the difference in their expected returns is given by:*

$$\eta(\tilde{\pi}) - \eta(\pi) = \frac{1}{1 - \gamma} \sum_s \rho_{\tilde{\pi}}(s) \sum_a \tilde{\pi}(a|s) A_\pi(s, a). \tag{4}$$

Theorem 2.1 implies that maximizing the expected advantage at every state guarantees performance improvement. However, the complex dependency of $\rho_{\tilde{\pi}}$ on the target policy $\tilde{\pi}$ makes direct optimization intractable. Schulman et al. (2015a) addressed it by deriving the following lower bound:

**Theorem 2.2.** *(Schulman et al., 2015a) Let $\epsilon = \max_{s,a} |A_\pi(s, a)|$ and $\beta = \frac{4\epsilon\gamma}{(1-\gamma)^2}$. The following inequality holds:*

$$\begin{aligned} \eta(\tilde{\pi}) - \eta(\pi) \geq &\frac{1}{1 - \gamma} \mathbb{E}_{s \sim \rho_\pi(\cdot), a \sim \tilde{\pi}(\cdot|s)} \left[ A_\pi(s, a) \right] \\ &- \beta \left[ D_{\mathrm{TV}}^{max}(\pi, \tilde{\pi}) \right]^2, \end{aligned} \tag{5}$$

*where $D_{\mathrm{TV}}^{\max}(\pi, \tilde{\pi}) = \max_s D_{\mathrm{TV}}(\pi(\cdot|s)\|\tilde{\pi}(\cdot|s))$ denotes the maximum Total Variation divergence.*

By applying importance sampling with $\tilde{\pi}(\cdot|s)$, we can rewrite the first term. Denote the surrogate objective $S(\tilde{\pi})$:

$$S(\tilde{\pi}) = \eta(\pi) + \frac{1}{1 - \gamma} \mathbb{E}_{s \sim \rho_\pi(\cdot), a \sim \pi(\cdot|s)} \left[ \frac{\tilde{\pi}(a|s)}{\pi(a|s)} A_\pi(s, a) \right]. \tag{6}$$

Substituting this into Eq. (5), we obtain:

$$\eta(\tilde{\pi}) \geq S(\tilde{\pi}) - \beta \left[ D_{\mathrm{TV}}^{max}(\pi, \tilde{\pi}) \right]^2. \tag{7}$$

Define the surrogate function $M(\tilde{\pi}) = S(\tilde{\pi}) - \beta [D_{\mathrm{TV}}^{max}(\pi, \tilde{\pi})]^2$. Notice that when $\tilde{\pi} = \pi$, we have $\eta(\tilde{\pi}) = M(\tilde{\pi}) = \eta(\pi)$. Consequently, if there exists a policy $\tilde{\pi}$ such that $M(\tilde{\pi}) > M(\pi)$, it strictly guarantees $\eta(\tilde{\pi}) > \eta(\pi)$. Thus, maximizing the surrogate $M(\tilde{\pi})$ results in monotonic policy improvement.

Since the TV divergence is difficult to optimize directly, employ Pinsker's inequality to derive a tractable lower bound using KL divergence:

$$\eta(\tilde{\pi}) - \eta(\pi) \geq S(\tilde{\pi}) - \frac{\beta}{2} D_{\mathrm{KL}}^{max}(\pi, \tilde{\pi}), \tag{8}$$

where $D_{\mathrm{KL}}^{\max}(\pi, \tilde{\pi}) = \max_s D_{\mathrm{KL}}(\pi(\cdot|s)\|\tilde{\pi}(\cdot|s))$.

It is a standard result in optimization that the unconstrained problem $\max_{\tilde{\pi}} S(\tilde{\pi}) - \frac{\beta}{2} D_{\mathrm{KL}}^{\max}(\pi, \tilde{\pi})$ is equivalent to the constrained problem (9), provided that $\beta$ is the optimal Lagrange multiplier corresponding to the threshold $\epsilon$:

$$\max_{\tilde{\pi}} S(\tilde{\pi}) \quad \text{s.t.} \quad D_{\mathrm{KL}}^{\max}(\pi, \tilde{\pi}) \leq \epsilon, \tag{9}$$

where the threshold $\epsilon$ depends on $\beta$. However, motivated by practical considerations such as implementation simplicity and the need for more permissive update steps, TRPO treats $\epsilon$ as a tunable hyperparameter rather than strictly deriving it from $\beta$, and replaces the maximum KL divergence with the average KL divergence $\mathbb{E}_{s \sim \rho_\pi(\cdot)}[D_{\mathrm{KL}}(\pi(\cdot|s)\|\tilde{\pi}(\cdot|s))]$.

## 3. Unified Theoretical Perspective

Before proposing our main algorithm, we first propose a novel paradigm for the design philosophy of Policy Gradient (PG) algorithms as a preliminary in this section.

### 3.1. Theoretical Foundation

We begin by establishing a theoretical foundation that simplifies the analysis of trust-region constraints and motivates our design paradigm. To this end, we present the following lower bound on policy performance.

**Theorem 3.1** (The Dual-Ratio Lower Bound). *Let $\eta(\tilde{\pi})$ denote the expected return of policy $\tilde{\pi}$ as defined in Eq. 1, and let $S(\tilde{\pi})$ be the surrogate objective defined in Eq. 6. The following bound holds:*

$$\begin{aligned} \eta(\tilde{\pi}) \geq &S(\tilde{\pi}) - \frac{\beta\alpha}{2} \max_{s,a} \left[ \ln \frac{\tilde{\pi}(a|s)}{\pi(a|s)} \right] \\ &- \frac{\beta(1-\alpha)}{2} \max_{s,a} \left[ \ln \frac{\pi(a|s)}{\tilde{\pi}(a|s)} \right], \end{aligned} \tag{10}$$

*where $\alpha \in [0, 1]$ is a manually tuned hyperparameter, and equality holds when $\tilde{\pi} = \pi$.*

*Proof.* See Appendix E. $\qquad\square$

The **Dual-Ratio Lower Bound** (Theorem 3.1) offers a crucial insight: a valid conservative surrogate can be constructed via a convex combination of forward and reverse probability ratios. Unlike bounds used in Theorem 2.2 and TRPO which depend on the expectation of divergence, Theorem 3.1 implies that the penalty is determined by the worst-case ratios. However, estimating these global maxima is computationally intractable. Consequently, instead of explicitly calculating the penalty, we reformulate the objective to strictly bound the ratios element-wise.

We now define a generalized objective function designed to enforce these constraints:

**Definition 3.2** (The Unified Ratio Objective). Let $r(s, a) = \frac{\tilde{\pi}(a|s)}{\pi(a|s)}$. We consider the extended real number line, denoted as $\overline{\mathbb{R}} = \mathbb{R} \cup \{-\infty, +\infty\}$. Let $f, g : \mathbb{R} \to \overline{\mathbb{R}}$ be extended real-valued shaping functions. We impose the following **Topological Constraints** to define a valid trust region:

1. **Identity Anchoring (Fixed Point Condition):** The mapping must preserve the identity transformation, i.e., $f(1) = g(1) = 1$.

2. **Trust Region Bounding (Geometric Enclosure):** The functions must topologically enclose the identity map such that $g(x) \geq x \geq f(x)$ for all $x \in \mathbb{R}$.

We define the **Unified Ratio Objective** $F(\tilde{\pi}; f, g)$ as:

$$F(\tilde{\pi}; f, g) = \mathbb{E}_{\substack{s \sim \rho_\pi(\cdot) \\ a \sim \pi(\cdot|s)}} \left[ \min\left( g(r) A_\pi(s, a), \, f(r) A_\pi(s, a) \right) \right]. \tag{11}$$

In particular, if the graphs of $f(x)$ and $g(x)$ are symmetric with respect to the point $(1, 1)$, $F$ will treat policy reinforcement and suppression symmetrically. This symmetrical structure is not only physically intuitive but also simplifies the theoretical analysis. We term this specific form the **Symmetric Unified Objective**, denoted as $F_s(\tilde{\pi}; f)$. This paper primarily focuses on this symmetric form.

**Corollary 3.3.** *For the current policy $\tilde{\pi} = \pi$, we have $F(\pi; f, g) = 0$.*

*Proof.* This follows directly from the Identity Anchoring condition in Definition 3.2 and the definition of the advantage function. $\qquad\square$

Finally, we provide a theoretical guarantee for this design paradigm. We show that by incorporating hard constraints into the shaping functions defined by our topological constraints, monotonic improvement is strictly ensured.

**Theorem 3.4** (The Monotonic Improvement Condition). *Let $\delta_C(\cdot)$ be the extended real-valued indicator function (0 if $x \in C$, $\infty$ otherwise). For any shaping functions $f$ and $g$ satisfying the topological constraints in Definition 3.2, there exist non-negative constants $\epsilon_l, \epsilon_u$ defining a feasible interval $I = [1 - \epsilon_l, 1 + \epsilon_u]$. If $\pi^*$ maximizes the constrained objective*

$$F(\pi^*; f_\delta, g_\delta) = \max_{\tilde{\pi}} F(\tilde{\pi}; f_\delta, g_\delta), \tag{12}$$

*where $f_\delta = f - \delta_I$ and $g_\delta = g + \delta_I$, then monotonic improvement is strictly guaranteed, i.e., $\eta(\pi^*) \geq \eta(\pi)$. Crucially, for ill-behaved shaping functions, the guarantee holds by forcing degeneration ($\epsilon_l, \epsilon_u \to 0$), reducing the update to a safe identity operation.*

*Proof.* See Appendix F. The proof utilizes the Dual-Ratio Lower Bound (Theorem 3.1) and indicator function properties. $\qquad\square$

## 3.2. Practical Design Paradigm

Directly solving the constrained optimization problem (e.g., Eq.15 in (Xie et al., 2025)) is often intractable or leads to prohibitively small updates in deep reinforcement learning (Schulman et al., 2015a). Consequently, established methods adopt a *relaxation strategy*: transforming hard constraints into surrogate objectives. For instance, TRPO approximates the penalty via a hard constraint (Schulman et al., 2015a), while PPO and SPO relax it into clipping or quadratic penalties (Xie et al., 2025). Following this paradigm, ANO proposes a *smooth relaxation* designed to approximate the ideal trust region while maintaining gradient continuity and robustness against outliers.

While Theorem 3.4 theoretically guarantees monotonic improvement by incorporating strict constraints ($\delta_C$) into the shaping functions, deriving the precise adaptive bounds $\epsilon_l$ and $\epsilon_u$ at every step is computationally intractable. Furthermore, even if these exact values were calculable, they often lead to excessively conservative updates (i.e., vanishingly small step sizes), a limitation also observed in the analysis of TRPO (Schulman et al., 2015a). Therefore, having reformulated the global trust region penalty into element-wise constraints via $f$ and $g$, we adopt a practical approach: treating $\epsilon_l$ and $\epsilon_u$ as tunable hyperparameters. This strategy offers two key advantages: (1) it circumvents the prohibitive computational burden, and (2) it enables more permissive and efficient policy updates.

In general, the optimal bounds $\epsilon_u$ and $\epsilon_l$ are not necessarily symmetric (Theorem 3.1 & 3.4). This explains why certain PPO implementations achieve better performance by setting distinct clipping thresholds for the upper and lower bounds of policy ratios. **However, for conceptual simplicity and standard practice, we assume $\epsilon_u = \epsilon_l = \epsilon$ in the remainder of this paper.** We demonstrate in Remark 3.5 that this symmetry assumption is not arbitrary; specifically, we show that there exists a mechanism where adjusting the hyperparameter $\alpha$ naturally yields symmetric constraints.

*Remark* 3.5 (Symmetric Bounds via $\alpha$-Adjustment). Consider a single-state MDP with $\mathcal{A} = \{a_1, a_2, a_3\}$, policy $\pi = [0.2, 0.7, 0.1]$, and advantages $A = [10, -2, -6]$. Let $\beta = 8$ and the base shaping functions be linear, $f(r) = g(r) = r$. By setting the convex combination coefficient $\alpha = 0.96$ in Theorem 3.1, the optimal constraints derived for maximizing the objective become perfectly symmetric with $\epsilon_u = \epsilon_l = 0.6$. See Appendix G for the detailed derivation.

Under the assumption of symmetric bounds ($\epsilon_u = \epsilon_l = \epsilon$) and symmetric shaping functions ($f$ and $g$ are symmetric about $(1, 1)$), the objective $F(\tilde{\pi}; f_\delta, g_\delta)$ reduces to the **Symmetric Unified Objective** $F_s(\tilde{\pi}; f_\delta)$. It allows us to focus exclusively on the design of a single shaping function $f$.

In practical implementations, we typically relax the constraints based on the sign of the advantage function: we remove the lower constraint for samples where $A(s, a) > 0$ (since we encourage the ratio to increase) and the upper constraint where $A(s, a) < 0$ (since we encourage the ratio to decrease). **Following this intuition, the constrained function $f_\delta$ can be simplified to a one-sided constraint form: $f_u(r) = f(r) - \delta_{[0,1+\epsilon]}(r)$.**

Furthermore, to encourage more aggressive exploration, we can soften the hard indicator penalty $\delta_{[0,1+\epsilon]}(r)$ into a continuous penalty or simply a saturation function. For instance, replacing the infinite penalty with a zero-gradient saturation $\mathbb{I}_{r>1+\epsilon}(f(r) - f(1+\epsilon))$, and using a linear base function $f(r) = r$, we obtain:

$$f_{\text{PPO}}(r) = r - \mathbb{I}_{r>1+\epsilon}(r - (1 + \epsilon)) = \min(r, 1 + \epsilon). \quad (13)$$

It is straightforward to verify that substituting $f_{\text{PPO}}$ into our symmetric objective $F_s(\tilde{\pi}; f_{\text{PPO}})$ recovers the standard PPO-Clip objective.

Alternatively, if we choose a quadratic base function, which naturally imposes a soft constraint by having a local maximum, we can uniquely determine its form by imposing three conditions: (1) $f(1) = 1$ and (2) $f(r) \leq r$, **both of which are derived directly from the consistency requirements in Definition 3.2**; and (3) the maximum is located at $1 + \epsilon$. This yields:

$$f_{\text{SPO}}(r) = -\frac{1}{2\epsilon}(r - 1 - \epsilon)^2 + \frac{\epsilon}{2} + 1. \quad (14)$$

Substituting this into $F_s(\tilde{\pi}; f_{\text{SPO}})$ recovers the objective function of SPO (Xie et al., 2025).

### 3.3. Principles for Ideal Shaping Functions

To guide the design of this *optimal relaxation*, we establish a set of Design Criteria for the shaping function. These criteria bridge the gap between rigorous optimization theory and practical stability.

Although PPO and SPO can be derived from our unified framework via specific relaxations, they exhibit notable limitations in practice when viewed through this theoretical lens. First, PPO employs a "hard clipping" mechanism (Eq. 13). While this enforces the trust region, it introduces a zero-gradient plateau for samples outside the clipping range. Consequently, valuable information from distributionally mismatched samples (outliers) is effectively discarded, leading to sample inefficiency (Wang et al., 2020). Second, while SPO introduces a quadratic penalty (Eq. 14) to ensure smoothness, its rigid shape leads to **linearly increasing gradients**. As we discuss below, this unboundedness violates the robustness requirements for handling outliers, creating stability risks.

To address these issues, we propose four design principles for an ideal shaping function $f(r)$. An effective shaping

function should possess the following independent properties:

1. **Global Differentiability:** Ideally, $f$ should be globally differentiable on $\mathbb{R}$. Given the symmetric construction of $g$ in the objective $F_s$, differentiability ensures that the optimization landscape is smooth. This property is essential for the convergence guarantees of first-order optimizers (Bottou et al., 2018), allowing for consistent gradient propagation without the instability introduced by non-differentiable points (kinks) typically found in hard-clipping objectives.

2. **Bounded Maximization Region:** The set of global maximizers of $f$ must be contained within a bounded interval. This property creates an implicit trust region. Even when the sample batch is insufficient to cover the entire state-action space, the geometric constraint ensures that the policy ratio is naturally confined within a safe region corresponding to $\epsilon$. Notably, PPO often fail to strictly enforce this confinement in practice, leading to unconstrained updates (Ilyas et al., 2018; Wang et al., 2020).

3. **Boundedness of (Sub)gradients:** Crucially, regardless of whether $f$ is smooth or non-smooth, its first-order information (gradients or subgradients) must be bounded. In the presence of extreme policy ratios (outliers), unbounded updates can lead to catastrophic divergence. Limiting gradient magnitude is a fundamental technique to prevent explosion (Pascanu et al., 2013; Zhang et al.). Furthermore, from the perspective of robust statistics, this acts as a safety mechanism ensuring a bounded influence function, a necessary condition for bias-robustness (Huber, 1992).

4. **Structural Simplicity:** The shaping function should adhere to the principle of parsimony. It should possess the simplest geometric structure necessary to enforce the trust region and robustness (e.g., maintaining a consistent curvature profile). Superfluous structural complexity such as unnecessary oscillations or intricate high-order derivatives should be avoided, as these can introduce spurious local optima and complicate the optimization landscape without theoretical justification.

## 4. Anchored Neighborhood Optimization

In this section, we present our proposed algorithm, **Anchored Neighborhood Optimization (ANO)**. Designed strictly following the principles established in Section 3.3, ANO serves as a practical realization of the ideal estimator derived from our unified theoretical framework. See Appendix D for pseudocode.

### 4.1. Rationale: The Gradient Dilemma

A critical challenge in on-policy RL is leveraging information from off-distribution samples ($r \neq 1$) safely. PPO's "hard clipping" frequently fails to contain the probability ratio within safe bounds (Ilyas et al., 2018; Wang et al., 2020) and suffers from sample inefficiency by discarding outliers. Unconstrained methods (e.g., SPO) expose the optimization to unbounded gradients. We argue that an ideal estimator must adopt distinct behaviors across two regimes:

**Regime 1: Moderate Deviation ($r \gtrsim 1 + \epsilon$).** Here, samples are reliable enough to provide a corrective signal. A Restoration Force (negative gradient) is essential to actively pull the policy back into the trust region. This pre-emptive correction prevents error accumulation (especially for optimizers obtaining momentum), ensuring the policy remains where the surrogate approximation is valid.

**Regime 2: Extreme Deviation ($r \gg 1 + \epsilon$).** When the ratio is excessively large, linearly increasing penalties become counterproductive due to two distinct risks: **(1) Gradient Interference:** A massive gradient from a single outlier can numerically overshadow the constructive signals from the majority of valid samples, effectively acting as high-magnitude noise. **(2) Uncontrolled Distributional Shift:** Due to the probability constraint $\sum \pi(a|s) = 1$, forcefully suppressing one probability via a massive gradient causes unpredictable inflation of others. This introduces severe variance and instability.

**Principle of Redescending Gradients.** Based on this analysis, we propose **Principle A (Redescending Negative Gradient):** Drawing inspiration from redescending M-estimators in robust statistics (Huber, 2011), the shaping function should exert a negative restoration force when $r > 1 + \epsilon$ to enforce constraints. Crucially, this force must gracefully decay to 0 as $r \to +\infty$ to prevent gradient interference and instability.

### 4.2. The Algorithm

As discussed previously, neither PPO nor SPO satisfies all the proposed design principles, as they only address a subset of the necessary constraints. Consequently, we propose ANO, which strictly adheres to all ideal properties. We define the ANO base kernel as:

$$\phi(z) := \ln(1 + 2^{-2z}) + \frac{4}{1 + 2^{-z}}. \qquad (15)$$

The shaping function $f_{\text{ANO}}(r)$ is defined by scaling and shifting $\phi(z)$ to satisfy the precisely anchoring requirements:

$$f_{\text{ANO}}(r) = \frac{45\epsilon}{32\ln 2}\left[\phi(-1) - \phi\left(\frac{r - 1 - \epsilon}{\epsilon}\right)\right] + 1. \quad (16)$$

Here, the constant terms involving $\phi(-1)$ ensure the conditions such as $f_{\text{ANO}}(1) = 1$ and $f'_{\text{ANO}}(1 + \epsilon) = 0$.

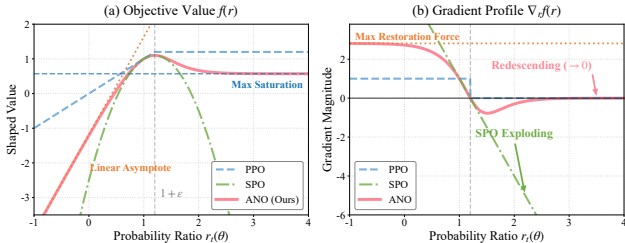

*Figure 2.* **Geometry of Shaping Functions.** A visual comparison of PPO (Hard Clip), SPO (Quadratic), and ANO (Redescending). ANO offers a unique profile that is smooth, bounded, and redescending, effectively reclaiming information from outliers without gradient explosion.

### 4.3. Properties and Analysis

In this section, we verify that ANO strictly adheres to the Design Criteria established in Section 3.3 and detailed proofs are provided in Appendix H. As summarized in Table 1, ANO exhibits superior geometric properties compared to existing baselines.

**1. $C^\infty$ Smoothness (Adhering to Principle 1).** Unlike PPO, which introduces non-differentiable "kinks" at $1 + \epsilon$, $f_{\text{ANO}}$ is constructed from elementary exponential and logarithmic functions. Consequently, it is not only globally differentiable but $C^\infty$ smooth. This smoothness ensures a well-defined Hessian everywhere, facilitating stable higher-order optimization dynamics (e.g., momentum accumulation and Hessian-based methods) and preventing the oscillation often observed around hard clipping boundaries.

**2. Implicit Trust Region (Adhering to Principle 2).** We rigorously analyze the critical points of $f_{\text{ANO}}$. By setting the derivative to zero, it is straightforward to verify that $f_{\text{ANO}}(r)$ possesses a **unique global maximum** at exactly $r = 1 + \epsilon$. See Appendix H.1 for proof. And PPO fails to bound its maximization region.

- For $r < 1 + \epsilon$, the function is strictly monotonically increasing ($f' > 0$).

- For $r > 1 + \epsilon$, the function is strictly monotonically decreasing ($f' < 0$).

This geometry creates an **implicit trust region**: maximizing the objective naturally encourages the policy ratio to gravitate towards the peak at $1 + \epsilon$, effectively bounding the policy update without requiring explicit brute-force clipping.

**3. Robustness via Redescending Gradients (Adhering to Principle 3 & A).** A defining feature of ANO is its handling of outliers, strictly following the rationale in Section 4.1. PPO forces gradients to zero too early, while SPO

allows gradients to grow unbounded. ANO strikes an optimal balance:

$$\lim_{r \to +\infty} f'_{\text{ANO}}(r) = 0, \quad \text{and} \quad \sup_r |f'_{\text{ANO}}(r)| < \infty. \quad (17)$$

Specifically, as $r \to +\infty$, the gradient gracefully decays to zero (the **Redescending property**), ensuring that extreme outliers do not destabilize the update via gradient interference. Conversely, for $r \to -\infty$, the gradient saturates to a constant factor ($\approx 45/16$), preserving a consistent restoration force. See Appendix H.4 for proof.

**4. Minimal Structural Complexity (Adhering to Principle 4).** Finally, we analyze the topological optimality of ANO. We argue that ANO achieves robustness with the **minimum theoretical structural complexity**.

**Proposition 4.1** (Necessity of Convexity Change). *Let $f$ be a continuous function on $\mathbb{R}$ satisfying **Bounded Maximization** and **Asymptotic Stability** (vanishing gradients). Then, $f$ cannot be globally concave on the tail interval; it must exhibit **at least one** change in convexity (inflection point).*

*Proof.* See Appendix I for details. □

**Optimality vs. Bell Curves.** One might suggest utilizing standard bell-shaped kernels (e.g., Gaussian or Welsch) from robust statistics. However, these functions typically fail the consistency constraint $f(r) \leq r$ (e.g., Gaussian exceeds $y = x$ at $r \to -\infty$) and require *two* inflection points. And ANO is perfectly designed to satisfy consistency on the left (via linearity) while providing the necessary convexity change on the right, achieving robustness with the theoretical lower bound of topological complexity (exactly one inflection point).

**Conclusion.** By satisfying all constraints with the theoretical lower bound of topological complexity, ANO represents the most parsimonious solution for robust policy optimization. With the theoretical properties established, we now turn to empirical validation to demonstrate ANO's performance in practice.

## 5. Experiments

We empirically validate Anchored Neighborhood Optimization (ANO) by addressing three Research Questions: **RQ 1 (Performance):** Does ANO consistently outperform baselines across diverse domains? **RQ 2 (Robustness):** Is ANO tolerant to aggressive hyperparameters? **RQ 3 (Mechanism):** How do ANO's theoretical principles manifest in practical training dynamics and exploration efficiency?

### 5.1. Traditional Reinforcement Learning Benchmarks

**Experimental Setup.** We evaluate ANO on the **MuJoCo** suite (Todorov et al., 2012), specifically focusing on 6 di-

*Table 1.* **Mathematical Properties of Policy Optimization Objectives.** ANO uniquely satisfies all principles. Note that PPO fails Principle 2 due to its unbounded plateau (Argmax region), while SPO fails Principle 3 due to unbounded gradients.

| PROPERTY | PPO | SPO | ANO (OURS) | PRINCIPLE |
|---|---|---|---|---|
| *Gradient Asymptotics (Tail Behavior)* | | | | |
| $r \to +\infty$ *(Outlier Handling)* | 0 (VANISHING) *(Info Loss)* | $\to -\infty$ (EXPLODING) *(Instability)* | $\to 0$ (**REDESCENDING**) *(Robustness)* | PRINCIPLE A |
| $r \to -\infty^*$ *(Restoration Force)* | 1 (CONSTANT) *(Linear)* | $\to +\infty$ (EXPLODING) *(Unsafe)* | $\to 45/16$ (**SATURATING**) *(Stable Force)* | EQ. (48) |
| GLOBAL BOUND $(\sup \|f'\| < \infty)$ | YES (✓) | No (×) | YES (✓) | PRINCIPLE 3 |
| MAXIMIZATION DOMAIN (ARGMAX REGION) | UNBOUNDED (×) *(Flat Plateau)* | BOUNDED (✓) *(Unique Peak)* | BOUNDED (✓) *(Unique Peak)* | PRINCIPLE 2 |
| TOPOLOGY (COMPLEXITY) | FLAT TAIL *(Trivial)* | GLOBAL CONVEX *(Rigid)* | SINGLE INFLECTION *(Minimal under Principle A)* | PRINCIPLE 4 |
| SMOOTHNESS (DIFFERENTIABILITY) | $C^0$ (KINKS) *(Noise)* | $\mathbf{C}^\infty$ *(Stable)* | $\mathbf{C}^\infty$ *(Stable)* | PRINCIPLE 1 |

*ANALYZED TO DETERMINE THE RIGHT-TAIL BEHAVIOR OF THE SYMMETRIC DUAL $g(r)$.

verse environments, and the full **Atari** benchmark containing 40 games (Bellemare et al., 2013). To ensure a rigorous and statistically meaningful comparison, We utilize *rliable* metrics (Agarwal et al., 2021) (e.g., IQM) aggregated across 5 seeds. We compare ANO against baselines including PPO, TRPO, and the recent state-of-the-art robust methods PAPO (Zhao et al., 2024) and SPO (Xie et al., 2025).

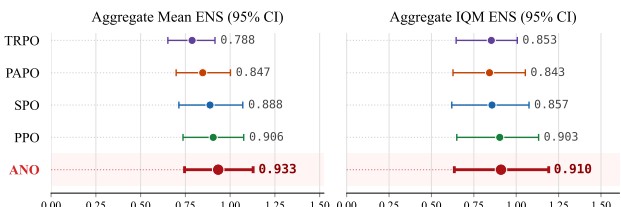

*Figure 3.* **Aggregate Performance on MuJoCo (6 Environments, 5 Seeds).** We report the Mean and Interquartile Mean (IQM) of expert normalized scores (ENS) with 95% stratified bootstrap confidence intervals. ANO (red) consistently achieves SOTA performance.

**Overall Performance (answering RQ 1).** As shown in Figure 3, ANO achieves SOTA on MuJoCo, surpassing the strongest baseline (PPO). Similarly, in the high-variance Atari domain (Figure 4), ANO maintains a significant lead in Human Normalized Score (HNS), effectively handling sparse rewards. The per-game learning curves and full score breakdowns are provided in Appendix C.

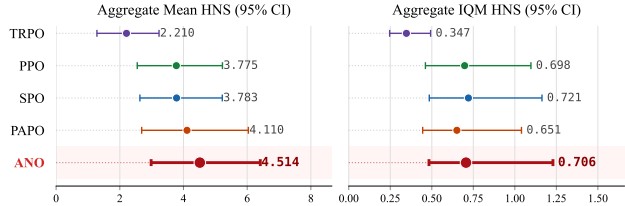

*Figure 4.* **Aggregate Performance on Atari (40 Environments, 5 Seeds).** We report the Mean and IQM of human normalized scores (HNS). ANO demonstrates superior stability and final performance compared to PPO and SPO.

**Robustness Analysis (answering RQ 2).** We stress-test robustness by increasing the learning rate to $1 \times 10^{-3}$. As illustrated in Figure 5, while PPO suffers catastrophic degradation ($-37.3\%$), ANO maintains performance ($-7.1\%$), confirming its intrinsic regularization effectively safeguards updates.

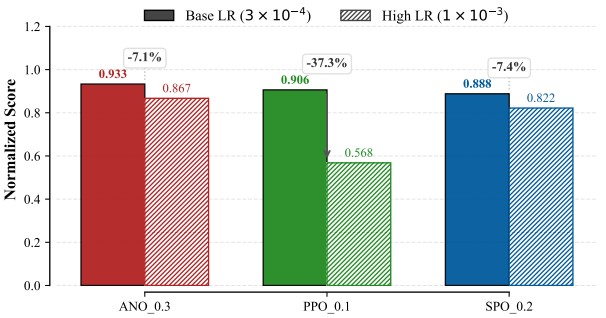

*Figure 5.* **Robustness Analysis on MuJoCo.** ANO demonstrates exceptional robustness under high learning rates (`1e-3`), whereas PPO suffers catastrophic performance degradation.

### 5.2. Large Language Model Fine-tuning

Beyond control tasks, we evaluate ANO on RLHF using the **TL;DR** summarization dataset (Stiennon et al., 2020). Following the `TRL` framework (von Werra et al., 2020), we fine-tune a `Pythia-1b-deduped` policy (Biderman et al., 2023) and compare it against PPO and GRPO (Shao et al., 2024).

**Win-Rate Analysis.** We conduct a large-scale head-to-head evaluation using **DeepSeek-V3** (Liu et al., 2024) as the judge. And we validate our strong baselines: PPO, SPO, and GRPO all exceed a 70% win rate against the SFT model at $T = 0$.

As shown in Figure 6, ANO demonstrates **consistent superiority across all temperatures**. Unlike standard methods that fluctuate, ANO maintains a robust win rate against PPO,

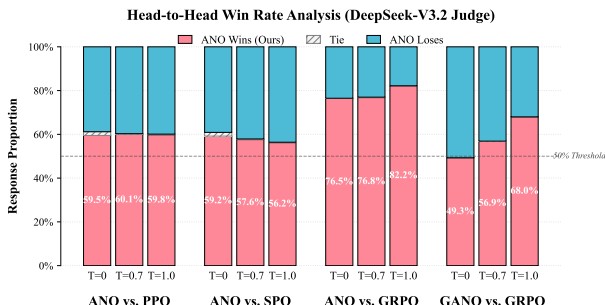

*Figure 6.* **Head-to-Head Win Rates.** ANO consistently outperforms PPO ($\sim 60\%$), SPO, and GRPO ($> 76\%$) across all sampling temperatures ($T \in \{0, 0.7, 1.0\} \times 1000$ samples).

ranging from **59.5%** ($T = 0$) to a peak of **60.1%** ($T = 0.7$). The margin is even more pronounced against GRPO, where ANO achieves up to **82.2%** win rate ($T = 1.0$). This confirms that ANO does not trade off diversity for quality; rather, it dominates the entire sampling spectrum.

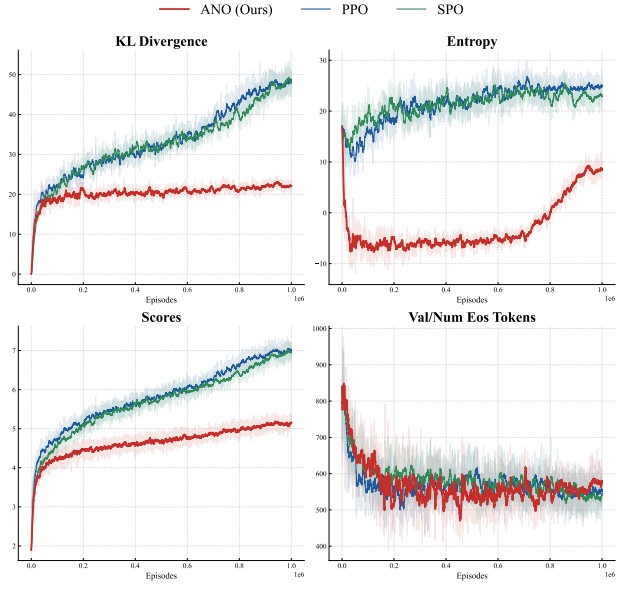

*Figure 7.* **Training Dynamics.** ANO maintains stable KL divergence and structured entropy.

**Mechanism Analysis (answering RQ 3).** Analyzing training dynamics (Figure 7) reveals:

- **Mitigating Goodhart's Law:** PPO achieves higher proxy rewards ($\sim 7.0$) than ANO ($\sim 5.2$) but loses head-to-head, confirming PPO suffers from reward hacking while ANO prioritizes alignment fidelity.

- **Semantic Stability (KL):** PPO and SPO both breach the trust region ($KL > 50$) but for opposite reasons: PPO suffers *drift* due to zero gradients on outliers, while SPO *destabilizes* the simplex via unbounded gradients. In contrast, ANO maintains stability ($KL \approx 20$) by applying a *redescending* force, effectively anchoring the policy to the knowledge manifold.

- **Structured Exploration (Entropy):** ANO exhibits a **"U-shaped" dynamic** (rapid pruning then stabilization). Addressing potential concerns of mode collapse during the entropy dip, we found an **intermediate checkpoint** achieved a $\sim 70\%$ win rate (vs. fully converged PPO) at $T = 0.7$ (and comparable $\sim 45\%$ at $T = 0$). This confirms the drop reflects the **filtering of low-quality tails**, concentrating mass on a high-quality core, rather than degenerate collapse.

**Computational Efficiency.** Theoretically, ANO involves transcendental operations ($\exp, \log$) which are strictly costlier than PPO's primitive clipping. However, this element-wise overhead is negligible relative to the dominant costs of neural network forward-backward passes and attention mechanisms. As shown in Table 2, ANO (61.72h) actually finished slightly faster than PPO (64.08h). We attribute this counter-intuitive result to normal hardware fluctuations and cluster load variability rather than algorithmic superiority. Fundamentally, both algorithms share the same time complexity $\mathcal{O}(N)$, confirming that ANO introduces little effective latency in practice.

*Table 2.* **Training Time.**

| ALGO | ANO (OURS) | PPO | SPO | GRPO | GANO (OURS) |
|---|---|---|---|---|---|
| TIME (H) | 61.72 | 64.08 | 60.68 | 14.00 | 7.36 |

**Remark on Compatibility.** It is worth noting that ANO's improvement (gradient shaping) is orthogonal to GRPO's structural innovation (group-based normalization without a value function). Consequently, ANO can be seamlessly integrated into the GRPO framework by simply replacing the clipped objective with the ANO shaping function. This suggests a promising avenue for future work to combine both strengths for even greater efficiency in LLM alignment.

## 6. Conclusion

In this work, we introduced the **Unified Trust Region Framework (UTF)**, a theoretical paradigm that resolves the tension between stability and efficiency in on-policy reinforcement learning. By rigorously establishing topological boundaries for monotonic improvement, UTF effectively **decouples theoretical safety from objective design**, empowering the community to explore novel shaping functions freely within a theoretically grounded "safe zone" without the burden of re-deriving convergence guarantees. Our instantiation, **ANO**, validates this framework by demonstrating great robustness across high-dimensional control and language modeling with negligible computational overhead. **Given its theoretical elegance and implementation simplicity, we advocate for ANO to supersede PPO as the new standard default for robust policy optimization.**

## Impact Statement

This paper focuses on advancing the theoretical and practical foundations of deep reinforcement learning through improved gradient estimation. The proposed algorithm, ANO, enhances training stability in complex environments, ranging from robotic control to language model fine-tuning. We believe this work primarily impacts the research community by offering a more robust tool for training agents. While more efficient optimization can theoretically accelerate both beneficial and harmful applications of AI, the core contribution of this work, stability, is fundamentally a safety feature. We encourage practitioners to apply this algorithm responsibly, particularly in safety-critical domains.

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

# Appendix

## A. Experimental Details

In this section, we provide the comprehensive implementation details, hyperparameter settings, and computational resources used in our experiments to ensure reproducibility.

### A.1. Implementation Frameworks

Our experiments are built upon established open-source libraries to ensure reproducibility and high efficiency:

- **Traditional RL (MuJoCo & Atari):** We adopt the high-quality single-file implementations from `CleanRL` (Huang et al., 2022). To maximize training throughput, we utilize `EnvPool` (Weng et al., 2022b) for massive parallel environment simulation, which significantly accelerates the wall-clock training time compared to standard `Gymnasium` (Towers et al., 2024) wrappers.

- **LLM Fine-tuning:** We use the `TRL` library (von Werra et al., 2020) integrated with `Accelerate` (Gugger et al., 2022) and `DeepSpeed` (Rasley et al., 2020) for efficient training. The reward model is a fine-tuned Pythia-1B scoring model, consistent with the standard CleanRL setup for efficient 1B-scale experiments.

For MuJoCo, we apply Expert Normalized Score (ENS):

$$\text{ENS} = \frac{\text{Agent Score} - \text{Random Score}}{\text{Expert Score} - \text{Random Score}}, \tag{18}$$

where the expert we used is TD3 (Fujimoto et al., 2018) just like Lee et al. (2025) and the TD3 scores are from Weng et al. (2022a) and random scores are from Lee et al. (2025).

For Atari, we apply Human Normalized Score (HNS):

$$\text{HNS} = \frac{\text{Agent Score} - \text{Random Score}}{\text{Human Score} - \text{Random Score}}, \tag{19}$$

where the human scores and random scores are from Mnih et al. (2015).

To ensure a strictly fair comparison between ANO, PPO, and GRPO, we aligned the **Global Batch Size** and **Total Training Episodes** across all algorithms. Table 3 details the specific configurations used for the TL;DR summarization task.

### A.2. Hyperparameter Settings for Traditional RL

For MuJoCo and Atari tasks, we adopted the standard hyperparameters recommended by *rliable* baselines. For PPO, SPO and ANO, we performed a grid search for the $\epsilon \in \{0.1, 0.2, 0.3\}$. For TRPO, we performed a grid search for the $\delta \in \{0.01, 0.02, 0.03\}$. For PAPO, we performed a grid search for the $\omega \in \{0.001, 0.005, 0.01\}$.

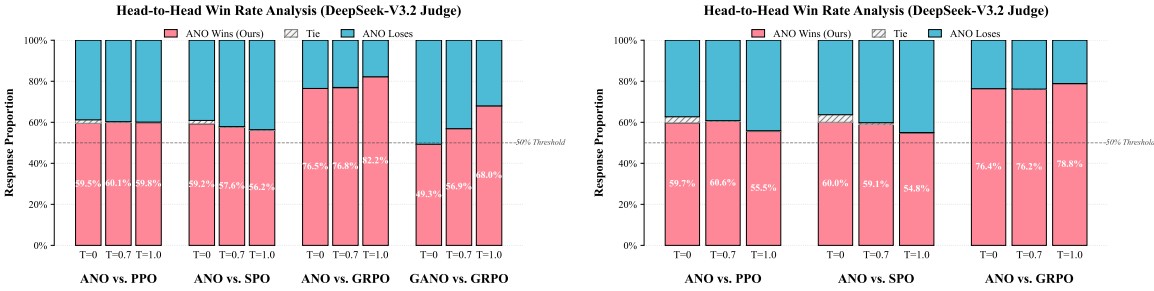

*Figure 8.* **Head-to-Head Win Rates.** The left and right panels display results for ANO with $\epsilon = 0.2$ and $\epsilon = 0.3$, respectively. ANO consistently outperforms all baselines across all sampling temperatures ($T \in \{0, 0.7, 1.0\}$) and $\epsilon$ settings, further demonstrating its robustness.

*Table 3.* **Hyperparameters for LLM Fine-tuning (TL;DR Task).** Note that we strictly align the Global Batch Size to 64 and Total Episodes to 1M across all methods to ensure fairness. These settings are recommended by (von Werra et al., 2020). Note that we also tried applying 3 update epochs for GRPO, but the performance is bad. And both 0.2 and 0.3 clip range make ANO outperform other methods.

| Hyperparameter | PPO/SPO | ANO (Ours) | GRPO | GANO (Ours) |
|---|---|---|---|---|
| Base Model | Pythia-1b-deduped | Pythia-1b-deduped | Pythia-1b-deduped | Pythia-1b-deduped |
| Dataset | TRL-Lib/tldr | TRL-Lib/tldr | TRL-Lib/tldr | TRL-Lib/tldr |
| Optimizer | AdamW | AdamW | AdamW | AdamW |
| Learning Rate | $3 \times 10^{-6}$ | $3 \times 10^{-6}$ | $3 \times 10^{-6}$ | $3 \times 10^{-6}$ |
| LR Scheduler | Cosine Decay | Cosine Decay | Cosine Decay | Cosine Decay |
| *Fairness Alignment Config* | | | | |
| Per-Device Batch Size | 8 | 8 | 16 | 16 |
| Gradient Accumulation | 8 | 8 | 4 | 4 |
| **Global Batch Size** | **64** | **64** | **64** | **64** |
| Total Training Steps | 15,625 | 15,625 | 15,625 | 15,625 |
| **Total Episodes Seen** | **1,000,000** | **1,000,000** | **1,000,000** | **1,000,000** |
| *Algorithm Specifics* | | | | |
| Num. Generations ($G$) | N/A | N/A | 4 | 4 |
| Update Epochs | 4 | 4 | 1 | 3 |
| KL Coefficient ($\beta$) | 0.05 | 0.05 | 0.05 | 0.05 |
| Clip Range ($\epsilon$) | 0.2 | 0.2/0.3 | 0.2 | 0.2 |

## B. Additional Robustness Analysis

In the main text, we demonstrated ANO's robustness to Learning Rate variations. Here, we further analyze the sensitivity of ANO to its specific hyperparameter: the neighborhood radius $\epsilon$.

As shown in Figure 9 and Figure 10, ANO maintains high performance across a wide range of $\epsilon$ values (0.1 to 0.3), indicating that the method is not brittle to hyperparameter tuning.

## C. Full Experimental Results

We provide the detailed breakdown of aggregate metrics for MuJoCo in Figure 9 and Atari in Figure 10, complementing the summary figures in the main text. And the performance of every task is shown in Figure 11 and Figure 12. We also show the full LLM fine-tuning results in Figure

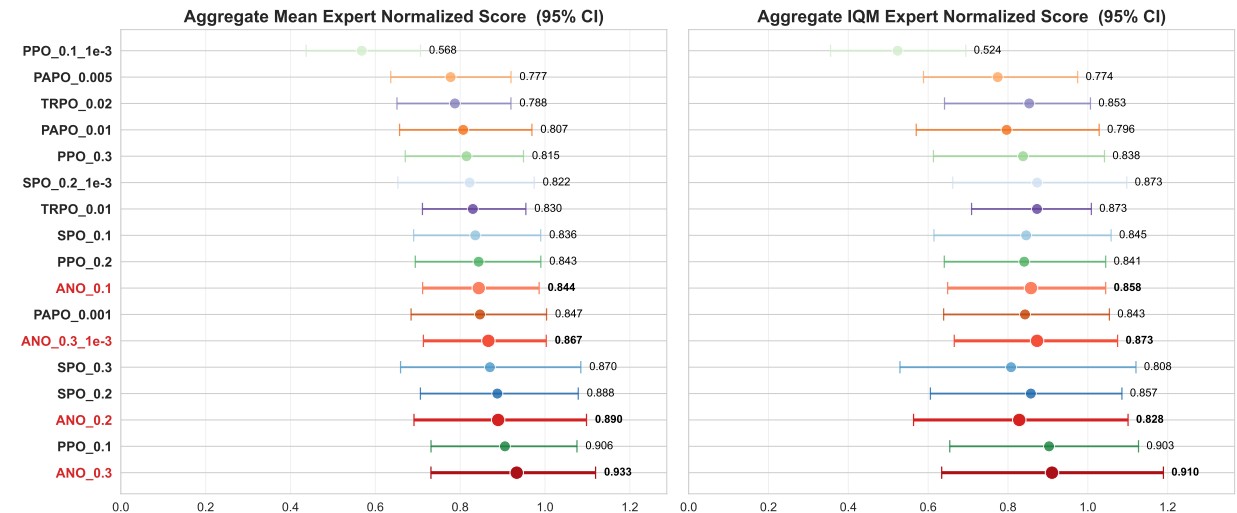

*Figure 9.* **Full Aggregate Performance on MuJoCo (**6 **Environments,** 5 **Seeds).** We report both IQM and Mean of Expert Normalized Scores (ENS). ANO demonstrates consistent superiority over PPO, TRPO, and SPO across both metrics.

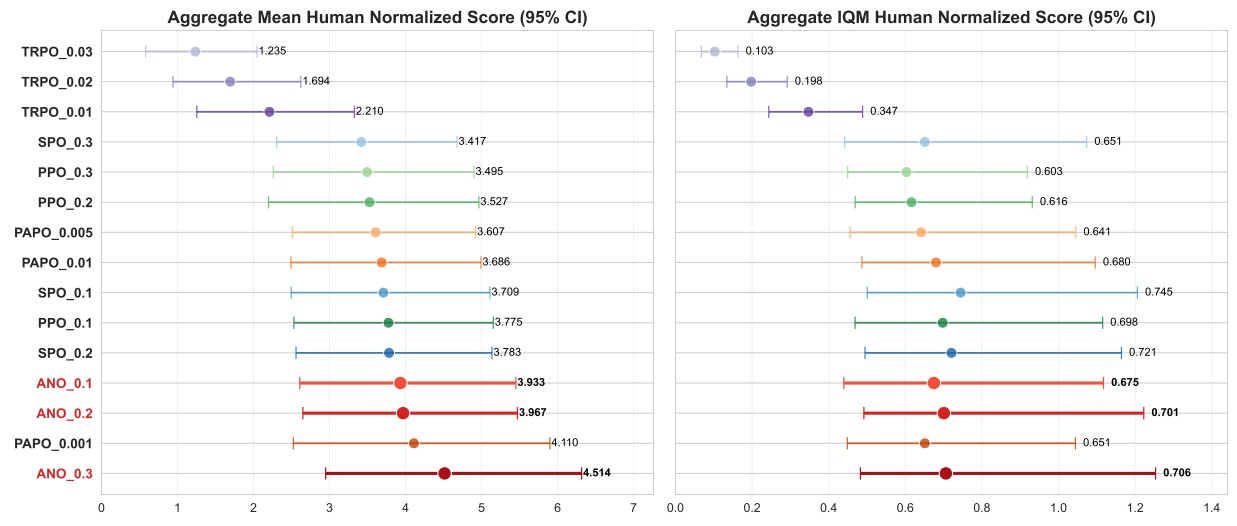

*Figure 10.* **Full Aggregate Performance on Atari (**40 **Environments,** 5 **Seeds).** We report both IQM and Mean of Human Normalized Scores (HNS). ANO achieves the highest aggregate mean scores and great IQM scores, proving its effectiveness in high-dimensional discrete control.

## D. Algorithm Implementation

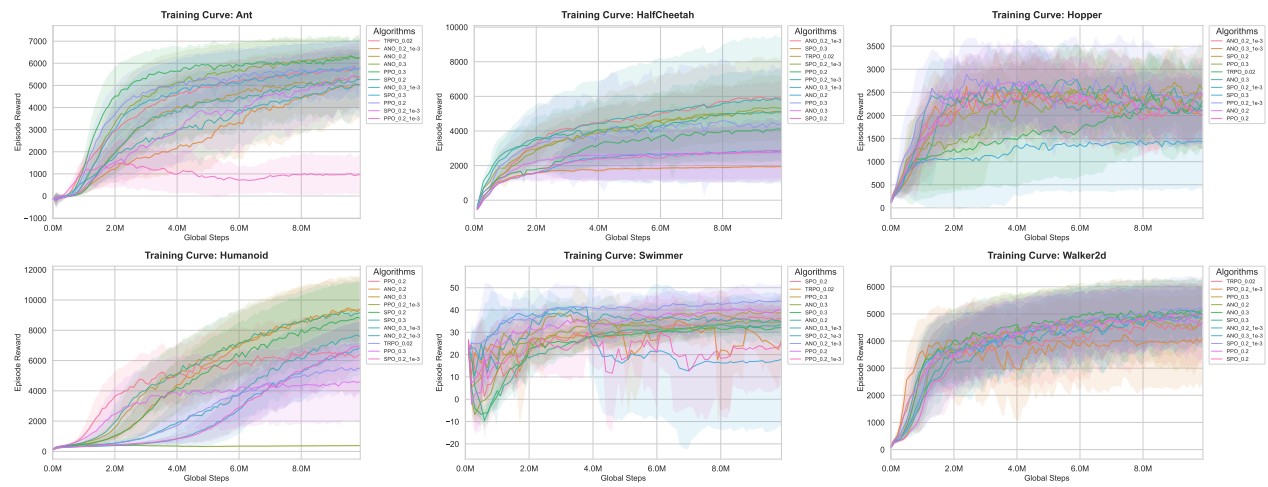

*Figure 11.* **Every Performance on Atari** (6 **Environments**, 5 **Seeds**).

---

**Algorithm 1** Training Procedure for Anchored Neighborhood Optimization (ANO)

---

1: **Require:** Initial parameters $\theta_0, \phi_0$; clipping/shaping threshold $\epsilon$; coefficients $\lambda_{val}, \lambda_{ent}$
2: **Hyperparameters:** Learning rates $\alpha_\theta, \alpha_\phi$; Discount $\gamma$, GAE parameter $\lambda$
3: **for** iteration $k = 0, 1, 2, \ldots$ **do**
4:    *# 1. Interaction & Data Collection*
5:    Sample trajectories $\mathcal{T} = \{(s_t, a_t, r_t)\}$ by running policy $\pi_{\theta_k}$ in the environment
6:    *# 2. Advantage Estimation*
7:    Compute generalized advantages $\hat{A}_t$ and value targets $\hat{R}_t$ using GAE $(\gamma, \lambda)$ based on $V_{\phi_k}$
8:    *# 3. Policy Snapshot*
9:    $\theta_{\text{old}} \leftarrow \theta_k$
10:   *# 4. Optimization Epochs*
11:   **for** epoch $m = 1, \ldots, M$ **do**
12:     Resample mini-batches from $\mathcal{T}$
13:     **for** each mini-batch $B$ **do**
14:       Calculate probability ratio $r_t(\theta) = \frac{\pi_\theta(a_t|s_t)}{\pi_{\theta_{\text{old}}}(a_t|s_t)}$
15:       *# Compute Unified Ratio Objective Loss*
16:       Define shaping function $f(r)$ using Eq. (16) {Apply ANO kernel}
17:       Derive symmetric dual $g(r) \leftarrow 2 - f(2 - r)$
18:       $\mathcal{L}_{\text{policy}} \leftarrow -\frac{1}{|B|} \sum_{t \in B} \min\left(g(r_t)\hat{A}_t, \ f(r_t)\hat{A}_t\right)$
19:       *# Auxiliary Losses*
20:       $\mathcal{L}_{\text{ent}} \leftarrow \frac{1}{|B|} \sum_{t \in B} \mathcal{H}(\pi_\theta(\cdot|s_t))$
21:       $\mathcal{L}_{\text{val}} \leftarrow \frac{1}{2|B|} \sum_{t \in B} (V_\phi(s_t) - \hat{R}_t)^2$
22:       *# Gradient Step*
23:       $\mathcal{L}_{\text{total}} \leftarrow \mathcal{L}_{\text{policy}} + \lambda_{\text{val}}\mathcal{L}_{\text{val}} - \lambda_{\text{ent}}\mathcal{L}_{\text{ent}}$
24:       Update $\theta, \phi$ w.r.t. $\nabla\mathcal{L}_{\text{total}}$ via optimizer
25:     **end for**
26:   **end for**
27:   $\theta_{k+1} \leftarrow \theta, \ \phi_{k+1} \leftarrow \phi$
28: **end for**

---

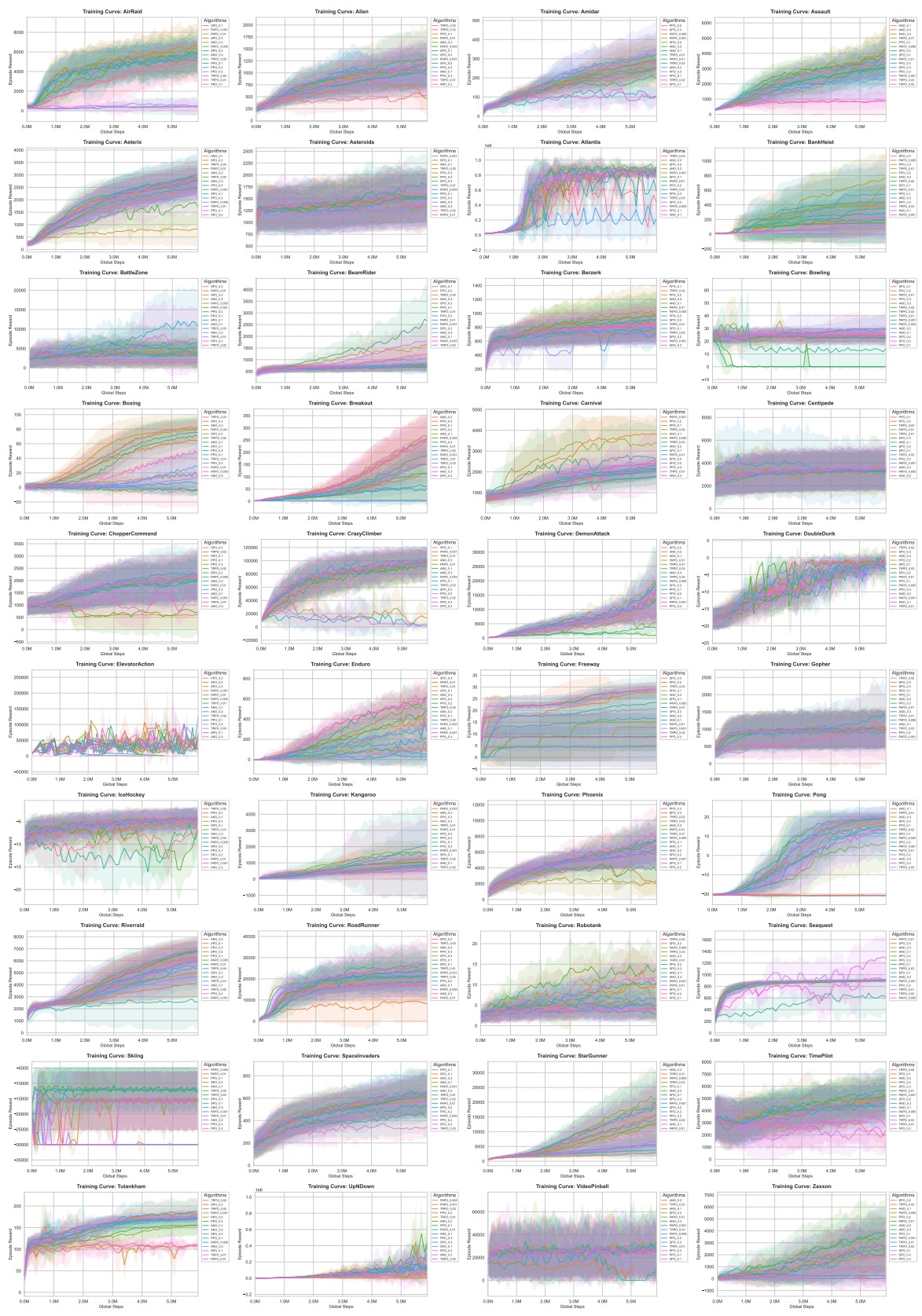

*Figure 12.* **Every Performance on Atari (**40 **Environments,** 5 **Seeds).**

# E. Proof of Theorem 3.1

**Theorem E.1.** *Let $\pi^*$ be the optimal policy derived from the definition of $\eta(\pi)$ in Eq. 1 and $S(\tilde{\pi})$ in Eq. 6:*

$$\eta(\tilde{\pi}) \geq S(\tilde{\pi}) - \frac{\beta\alpha}{2} \max_{s,a} \left[ \ln \frac{\tilde{\pi}(a|s)}{\pi(a|s)} \right] - \frac{\beta(1-\alpha)}{2} \max_{s,a} \left[ \ln \frac{\pi(a|s)}{\tilde{\pi}(a|s)} \right], \tag{20}$$

*where $\alpha \in [0,1]$ is a hyperparameter.*

*Proof.* From Theorem 2.2, we have

$$\eta(\tilde{\pi}) \geq S(\tilde{\pi}) - \beta \left[ D_{\mathrm{TV}}^{max}(\pi, \tilde{\pi}) \right]^2. \tag{21}$$

Suppose $a \in [0,1]$, and then we have

$$\eta(\tilde{\pi}) \geq S(\tilde{\pi}) - \alpha\beta \left[ D_{\mathrm{TV}}^{max}(\pi, \tilde{\pi}) \right]^2 - (1-\alpha)\beta \left[ D_{\mathrm{TV}}^{max}(\pi, \tilde{\pi}) \right]^2$$

$$= S(\tilde{\pi}) - \alpha\beta \left[ D_{\mathrm{TV}}^{max}(\tilde{\pi}, \pi) \right]^2 - (1-\alpha)\beta \left[ D_{\mathrm{TV}}^{max}(\pi, \tilde{\pi}) \right]^2$$

$$\geq S(\tilde{\pi}) - \frac{\alpha\beta}{2} \left[ D_{KL}^{max}(\tilde{\pi}, \pi) \right] - \frac{(1-\alpha)\beta}{2} \left[ D_{KL}^{max}(\pi, \tilde{\pi}) \right]$$

(By Pinsker's inequality)

$$= S(\tilde{\pi}) - \frac{\alpha\beta}{2} \max_s \mathbb{E}_{a\sim\tilde{\pi}(\cdot|s)} \left[ \ln \frac{\tilde{\pi}(a|s)}{\pi(a|s)} \right] - \frac{(1-\alpha)\beta}{2} \max_s \mathbb{E}_{a\sim\pi(\cdot|s)} \left[ \ln \frac{\pi(a|s)}{\tilde{\pi}(a|s)} \right]$$

$$\geq S(\tilde{\pi}) - \frac{\alpha\beta}{2} \max_{s,a} \left[ \ln \frac{\tilde{\pi}(a|s)}{\pi(a|s)} \right] - \frac{(1-\alpha)\beta}{2} \max_{s,a} \left[ \ln \frac{\pi(a|s)}{\tilde{\pi}(a|s)} \right] \tag{22}$$

$\square$

# F. Proof of Theorem 3.4

**Theorem F.1.** *For any $f$ and $g$ in Definition 3.2, there exists non-negative $\epsilon_l$ and $\epsilon_u$ such that for $F(\pi^*; f_\delta, g_\delta) = \max_{\tilde{\pi}} F(\tilde{\pi}; f_\delta, g_\delta)$ where $f_\delta = f - \delta_{[1-\epsilon_l, 1+\epsilon_u]}$ and $g_\delta = g + \delta_{[1-\epsilon_l, 1+\epsilon_u]}$, we have $\eta(\pi^*) \geq \eta(\pi)$. The equality holds iff $F(\pi^*; f_\delta, g_\delta) = F(\pi; f_\delta, g_\delta)$.*

*Proof.* From Theorem 3.1, we have

$$\eta(\tilde{\pi}) \geq S(\tilde{\pi}) - \frac{\alpha\beta}{2} \max_{s,a} \left[ \ln \frac{\tilde{\pi}(a|s)}{\pi(a|s)} \right] - \frac{(1-\alpha)\beta}{2} \max_{s,a} \left[ \ln \frac{\pi(a|s)}{\tilde{\pi}(a|s)} \right]$$

$$= \eta(\pi) + \frac{1}{1-\gamma} \mathbb{E}_{\substack{s\sim\rho_\pi(\cdot) \\ a\sim\pi(\cdot|s)}} \left[ \min \left( rA_\pi(s,a),\, rA_\pi(s,a) \right) \right] - \frac{\alpha\beta}{2} \max_{s,a} \left[ \ln \frac{\tilde{\pi}(a|s)}{\pi(a|s)} \right] - \frac{(1-\alpha)\beta}{2} \max_{s,a} \left[ \ln \frac{\pi(a|s)}{\tilde{\pi}(a|s)} \right]$$

$$= \eta(\pi) + \frac{1}{1-\gamma} \mathbb{E}_{\substack{s\sim\rho_\pi(\cdot) \\ a\sim\pi(\cdot|s)}} \left[ \mathbb{I}_{A_\pi(s,a)<0} rA_\pi(s,a) + \mathbb{I}_{A_\pi(s,a)>0} rA_\pi(s,a) \right]$$

$$- \frac{\alpha\beta}{2} \max_{s,a} \left[ \ln \frac{\tilde{\pi}(a|s)}{\pi(a|s)} \right] - \frac{(1-\alpha)\beta}{2} \max_{s,a} \left[ \ln \frac{\pi(a|s)}{\tilde{\pi}(a|s)} \right]$$

$$\geq \eta(\pi) + \frac{1}{1-\gamma} \mathbb{E}_{\substack{s\sim\rho_\pi(\cdot) \\ a\sim\pi(\cdot|s)}} \left[ \mathbb{I}_{A_\pi(s,a)<0} g(r)A_\pi(s,a) + \mathbb{I}_{A_\pi(s,a)>0} f(r)A_\pi(s,a) \right]$$

$$- \frac{\alpha\beta}{2} \max_{s,a} \left[ \ln \frac{\tilde{\pi}(a|s)}{\pi(a|s)} \right] - \frac{(1-\alpha)\beta}{2} \max_{s,a} \left[ \ln \frac{\pi(a|s)}{\tilde{\pi}(a|s)} \right]$$

$$\left( \text{Because } g(r) \geq r \geq f(r) \right)$$

$$\geq \eta(\pi) + \frac{1}{1-\gamma} \mathbb{E}_{\substack{s\sim\rho_\pi(\cdot) \\ a\sim\pi(\cdot|s)}} \left[ \min \left( g(r)A_\pi(s,a),\, f(r)A_\pi(s,a) \right) \right]$$

$$- \frac{\alpha\beta}{2} \max_{s,a} \left[ \ln \frac{\tilde{\pi}(a|s)}{\pi(a|s)} \right] - \frac{(1-\alpha)\beta}{2} \max_{s,a} \left[ \ln \frac{\pi(a|s)}{\tilde{\pi}(a|s)} \right]$$

$$\geq \underbrace{\eta(\pi) + \frac{1}{1-\gamma} F(\tilde{\pi}; f, g) - \frac{\alpha\beta}{2} \max_{s,a} \left[\ln \frac{\tilde{\pi}(a|s)}{\pi(a|s)}\right] - \frac{(1-\alpha)\beta}{2} \max_{s,a} \left[\ln \frac{\pi(a|s)}{\tilde{\pi}(a|s)}\right]}_{M_F(\tilde{\pi})}. \tag{23}$$

According to Corollary 3.3, we have that $M_F(\pi) = \eta(\pi)$.

And according to a standard result in optimization, the unconstrained problem $\max_{\tilde{\pi}} M_F(\tilde{\pi})$ is equivalent to the constrained problem:

$$\max_{\tilde{\pi}} F(\tilde{\pi}; f, g) \tag{24}$$

$$\text{s.t.} \begin{cases} \max_{s,a}\left[\ln \frac{\tilde{\pi}(a|s)}{\pi(a|s)}\right] \leq \epsilon_1, \\ \max_{s,a}\left[\ln \frac{\pi(a|s)}{\tilde{\pi}(a|s)}\right] \leq \epsilon_2. \end{cases} \tag{25}$$

Obviously, both $\epsilon_l$ and $\epsilon_u$ are non-negative. And when $\alpha = 0$, $\epsilon_1 = \infty$ while $\alpha = 1$, $\epsilon_2 = \infty$.

And the optimization can be rewritten as follows:

$$\max_{\tilde{\pi}} F(\tilde{\pi}; f, g) \tag{26}$$

$$\text{s.t.} \begin{cases} \max_{s,a} \frac{\tilde{\pi}(a|s)}{\pi(a|s)} \leq 1 + \epsilon_u, \\ \min_{s,a} \frac{\tilde{\pi}(a|s)}{\pi(a|s)} \geq 1 - \epsilon_l, \end{cases} \tag{27}$$

where $\epsilon_u = e^{\epsilon_1} - 1$ and $\epsilon_l = 1 - e^{-\epsilon_2}$.

Let $\pi^*$ be the optimal solution to this optimization problem. We have

$$\pi^* = \operatorname*{argmax}_{\tilde{\pi}} F(\tilde{\pi}; f, g) - \delta_{[1-\epsilon_l, 1+\epsilon_u]}(\max_r r) - \delta_{[1-\epsilon_l, 1+\epsilon_u]}(\min_r r)$$

$$= \operatorname*{argmax}_{\tilde{\pi}} F(\tilde{\pi}; f, g) - \sum_s \sum_a \rho_\pi(s) p_\pi(a|s) \left(\delta_{[1-\epsilon_l, 1+\epsilon_u]}(\max_r r) + \delta_{[1-\epsilon_l, 1+\epsilon_u]}(\min_r r)\right)$$

$$\left(\text{Note that } \beta\delta_C(\cdot) \equiv \delta_C(\cdot) \text{ for any } \beta > 0\right)$$

$$= \operatorname*{argmax}_{\tilde{\pi}} \left[\sum_s \sum_a \rho_\pi(s) p_\pi(a|s) \min\left(g(r) A_\pi(s,a), f(r) A_\pi(s,a)\right) - \sum_s \sum_a \rho_\pi(s) p_\pi(a|s)\left(\delta_{[1-\epsilon_l, 1+\epsilon_u]}(r)\right)\right]$$

$$= \operatorname*{argmax}_{\tilde{\pi}} \left[\mathbb{E}_{\substack{s \sim \rho_\pi(\cdot) \\ a \sim \pi(\cdot|s)}} \min\left(g(r) A_\pi(s,a), f(r) A_\pi(s,a)\right) - \mathbb{E}_{\substack{s \sim \rho_\pi(\cdot) \\ a \sim \pi(\cdot|s)}} \left(\delta_{[1-\epsilon_l, 1+\epsilon_u]}(r)\right)\right]$$

$$= \operatorname*{argmax}_{\tilde{\pi}} \left[\mathbb{E}_{\substack{s \sim \rho_\pi(\cdot) \\ a \sim \pi(\cdot|s)}} \min\left(g(r) A_\pi(s,a), f(r) A_\pi(s,a)\right)\right.$$

$$\left. - \mathbb{E}_{\substack{s \sim \rho_\pi(\cdot) \\ a \sim \pi(\cdot|s)}} A_\pi(a|s) \mathbb{I}_{A_\pi(a|s)>0}\left(\delta_{[1-\epsilon_l, 1+\epsilon_u]}(r)\right) + \mathbb{E}_{\substack{s \sim \rho_\pi(\cdot) \\ a \sim \pi(\cdot|s)}} A_\pi(a|s) \mathbb{I}_{A_\pi(a|s)<0}\left(\delta_{[1-\epsilon_l, 1+\epsilon_u]}(r)\right)\right]$$

$$= \operatorname*{argmax}_{\tilde{\pi}} \left[\mathbb{E}_{\substack{s \sim \rho_\pi(\cdot) \\ a \sim \pi(\cdot|s)}} \min\left(\left(g(r) + \delta_{[1-\epsilon_l, 1+\epsilon_u]}(r)\right) A_\pi(s,a), \left(f(r) - \delta_{[1-\epsilon_l, 1+\epsilon_u]}(r)\right) A_\pi(s,a)\right)\right]$$

$$= \operatorname*{argmax}_{\tilde{\pi}} \left[\mathbb{E}_{\substack{s \sim \rho_\pi(\cdot) \\ a \sim \pi(\cdot|s)}} \min\left(g_\delta(r) A_\pi(s,a), f_\delta(r) A_\pi(s,a)\right)\right]$$

$$= \operatorname*{argmax}_{\tilde{\pi}} F(\tilde{\pi}; f_\delta, g_\delta). \tag{28}$$

Degenerate case: It is easy to show that when $f(r) = 1 - \delta_{\{1\}}$ and $g(r) = 1 + \delta_{\{1\}}$, $\epsilon_u = 0$ and $\epsilon_l = 0$. $\qquad\square$

# G. Proof of Remark 3.5

We verify the calculation for the specific MDP instance: $\pi = [0.2, 0.7, 0.1]$, $A = [10, -2, -6]$, and $\beta = 8$. We seek to find $\alpha$ such that the trust region bounds are symmetric ($\epsilon_u = \epsilon_l = \epsilon = 0.6$).

*Proof.* The optimization problem with the penalty term is:

$$\max_{\tilde{\pi}} \sum_a \tilde{\pi}(a|s)A(a|s) - \frac{\beta\alpha}{2}\max_a \ln\frac{\tilde{\pi}(a)}{\pi(a)} - \frac{\beta(1-\alpha)}{2}\max_a \ln\frac{\pi(a)}{\tilde{\pi}(a)} - \lambda\left(\sum \tilde{\pi} - 1\right) \tag{29}$$

Assuming the optimal policy $\tilde{\pi}$ hits the upper bound at $a_1$ and the lower bound at $a_3$, while $a_2$ remains in the interior (unconstrained relative to the trust region bounds but satisfying the probability simplex).

The first-order optimality condition (setting gradients to zero) gives:

$$\frac{\partial\mathcal{L}}{\partial\tilde{\pi}_1} = 10 - \frac{4\alpha}{\tilde{\pi}_1} - \lambda = 0 \tag{30}$$

$$\frac{\partial\mathcal{L}}{\partial\tilde{\pi}_2} = -2 - 0 - \lambda = 0 \implies \lambda = -\mathbf{2} \tag{31}$$

$$\frac{\partial\mathcal{L}}{\partial\tilde{\pi}_3} = -6 + \frac{4(1-\alpha)}{\tilde{\pi}_3} - \lambda = 0 \tag{32}$$

Substituting $\lambda = -2$ into the equations for $a_1$ and $a_3$: 1. For $a_1$ (Upper Bound: $\tilde{\pi}_1 = \pi_1(1+\epsilon)$):

$$10 - \frac{4\alpha}{0.2(1+0.6)} + 2 = 0 \implies 12 = \frac{4\alpha}{0.32} \implies \alpha = 0.96$$

2. For $a_3$ (Lower Bound: $\tilde{\pi}_3 = \pi_3(1-\epsilon)$):

$$-6 + \frac{4(1-\alpha)}{0.1(1-0.6)} + 2 = 0 \implies -4 + \frac{4(1-\alpha)}{0.04} = 0 \implies 1-\alpha = 0.04 \implies \alpha = 0.96$$

Both conditions consistently yield $\alpha = 0.96$. Thus, the derivation is exact. $\square$

# H. Proofs and Derivations for ANO

Recall the definition of the base kernel:

$$\phi(z) := \ln(1 + 2^{-2z}) + \frac{4}{1 + 2^{-z}}. \tag{33}$$

The shaping function $f_{\text{ANO}}(r)$ is defined as:

$$f_{\text{ANO}}(r) = \frac{45\epsilon}{32\ln 2}\left[\phi(-1) - \phi\left(\frac{r-1-\epsilon}{\epsilon}\right)\right] + 1. \tag{34}$$

Here, the constant term involving $\phi(-1)$ ensures the anchoring condition $f_{\text{ANO}}(1) = 1$.

## H.1. Proof of Unique Maximum Point

*Proof.* Let $z(r) = \frac{r-1-\epsilon}{\epsilon}$. By applying the chain rule, we compute the derivative of $f_{\text{ANO}}(r)$:

$$f'_{\text{ANO}}(r) = -\frac{45\epsilon}{32\ln 2} \cdot \phi'(z) \cdot \frac{dz}{dr}$$

$$= -\frac{45\epsilon}{32\ln 2}\left[\frac{-2\ln 2 \cdot 2^{-2z}}{1 + 2^{-2z}} + \frac{-4\ln 2 \cdot 2^{-z} \cdot (-1)}{(1 + 2^{-z})^2}\right] \cdot \frac{1}{\epsilon}$$

$$= \frac{45}{32} \left[ \frac{2 \cdot 2^{-2z}}{1 + 2^{-2z}} - \frac{4 \cdot 2^{-z}}{(1 + 2^{-z})^2} \right] \tag{35}$$

$$= \frac{45}{16} \left[ \frac{2^{-2z}}{1 + 2^{-2z}} - \frac{2 \cdot 2^{-z}}{(1 + 2^{-z})^2} \right]$$

$$= \frac{45}{16} \left[ \frac{2^{-2z}(1 + 2^{-z})^2 - 2 \cdot 2^{-z}(1 + 2^{-2z})}{(1 + 2^{-2z})(1 + 2^{-z})^2} \right]$$

$$= \frac{45}{16} \left[ \frac{2^{-2z}(1 + 2 \cdot 2^{-z} + 2^{-2z}) - (2 \cdot 2^{-z} + 2 \cdot 2^{-3z})}{(1 + 2^{-2z})(1 + 2^{-z})^2} \right]$$

$$= \frac{45}{16} \left[ \frac{2^{-2z} + 2 \cdot 2^{-3z} + 2^{-4z} - 2 \cdot 2^{-z} - 2 \cdot 2^{-3z}}{(1 + 2^{-2z})(1 + 2^{-z})^2} \right]$$

$$= \frac{45}{16} \left[ \frac{2^{-4z} + 2^{-2z} - 2 \cdot 2^{-z}}{(1 + 2^{-2z})(1 + 2^{-z})^2} \right]$$

$$= \frac{45}{16} \underbrace{\left[ \frac{2^{-3z} + 2^{-2z} + 2^{-z+1}}{(1 + 2^{-2z})(1 + 2^{-z})^2} \right]}_{>0 \text{ for all } z \in \mathbb{R}} (2^{-z} - 1). \tag{36}$$

Since the fractional term in brackets consists solely of positive exponential terms, it is strictly positive. Therefore, the sign of the derivative is determined uniquely by the term $(2^{-z} - 1)$.

- $f'_{\text{ANO}}(r) = 0 \iff 2^{-z} = 1 \iff z = 0 \iff r = 1 + \epsilon$.

- $f'_{\text{ANO}}(r) > 0 \iff 2^{-z} > 1 \iff z < 0 \iff r < 1 + \epsilon$.

- $f'_{\text{ANO}}(r) < 0 \iff 2^{-z} < 1 \iff z > 0 \iff r > 1 + \epsilon$.

Thus, $r = 1 + \epsilon$ is the unique global maximum point. $\square$

### H.2. Proof of the Asymptotic Behavior

Recalling the definition of ANO objective from Eq. 16:

$$f_{\text{ANO}}(r) = C\left[\phi(-1) - \phi(z)\right] + 1, \quad \text{where } C = \frac{45\epsilon}{32 \ln 2}, \; z = \frac{r - 1 - \epsilon}{\epsilon}. \tag{37}$$

The base kernel is defined as $\phi(z) = \ln(1 + 2^{-2z}) + \frac{4}{1 + 2^{-z}}$. We analyze the asymptotic behavior in two limits.

**Right Tail ($r \to +\infty$).** As $r \to +\infty$, we have $z \to +\infty$. In this limit, terms $2^{-z}$ and $2^{-2z}$ vanish:

$$\lim_{z \to +\infty} \phi(z) = \ln(1 + 0) + \frac{4}{1 + 0} = 4. \tag{38}$$

Substituting this back into $f_{\text{ANO}}(r)$, we obtain the **Constant Saturation Asymptote**:

$$\lim_{r \to +\infty} f_{\text{ANO}}(r) = C\left[\phi(-1) - 4\right] + 1. \tag{39}$$

This confirms that ANO is bounded for outliers in the maximization direction.

**Left Tail ($r \to -\infty$).** As $r \to -\infty$, we have $z \to -\infty$. Let $z = -u$ where $u \to +\infty$. The kernel behaves as:

$$\phi(z) = \ln(1 + 2^{2u}) + \frac{4}{1 + 2^u} \tag{40}$$

$$\approx \ln(2^{2u}) + 0 \quad (\text{since } 2^u \gg 1) \tag{41}$$

$$= 2u \ln 2 = -2z \ln 2. \tag{42}$$

Substituting the approximation $\phi(z) \approx -2z \ln 2$ into $f_{\text{ANO}}(r)$:

$$f_{\text{ANO}}(r) \approx C\phi(-1) - C(-2z \ln 2) + 1 \tag{43}$$

$$= C\phi(-1) + 2C \ln 2 \left( \frac{r - 1 - \epsilon}{\epsilon} \right) + 1. \tag{44}$$

Focusing on the slope (coefficient of $r$):

$$\text{Slope} = \frac{2C \ln 2}{\epsilon} = \frac{2 \ln 2}{\epsilon} \cdot \frac{45\epsilon}{32 \ln 2} = \frac{45}{16}. \tag{45}$$

Thus, the **Linear Asymptote** as $r \to -\infty$ is given by:

$$f_{\text{ANO}}(r) \sim \frac{45}{16} r + \left( C\phi(-1) - \frac{45(1 + \epsilon)}{16} + 1 \right). \tag{46}$$

This explicitly proves that the restoration force saturates at a constant gradient of $45/16$.

### H.3. Proof of Unique Inflection Point

*Proof.* To analyze the convexity, we examine the roots of the second derivative. Let $x(r) = 2^{-z(r)}$. Since $r \mapsto z$ is linear and $z \mapsto 2^{-z}$ is monotonic, the mapping from $r \in \mathbb{R}$ to $x \in (0, +\infty)$ is strictly monotonic (specifically, decreasing).

Referring to the simplified form of the gradient in Eq. (35), we define an auxiliary function $g(x)$ proportional to $f'_{\text{ANO}}(r)$:

$$g(x) = \frac{x^2}{1 + x^2} - \frac{2x}{(1 + x)^2}. \tag{47}$$

The inflection points of $f_{\text{ANO}}$ correspond to the critical points of the gradient w.r.t $r$. Since $\frac{df'}{dr} = \frac{dg}{dx}\frac{dx}{dr}$ and $\frac{dx}{dr} \neq 0$, it suffices to find the roots of $g'(x)$. Differentiating $g(x)$ with respect to $x$:

$$\begin{aligned}
g'(x) &= \frac{d}{dx}\left( \frac{x^2}{1 + x^2} \right) - 2\frac{d}{dx}\left( \frac{x}{(1 + x)^2} \right) \\
&= \frac{2x(1 + x^2) - x^2(2x)}{(1 + x^2)^2} - 2\left[ \frac{1 \cdot (1 + x)^2 - x \cdot 2(1 + x)}{(1 + x)^4} \right] \\
&= \frac{2x}{(1 + x^2)^2} - 2\left[ \frac{(1 + x) - 2x}{(1 + x)^3} \right] \\
&= \frac{2x(1 + x)^3 - 2(1 - x)(1 + x^2)^2}{(1 + x^2)^2(1 + x)^3} \\
&= \frac{2(x^5 + 5x^3 + x^2 + 2x - 1)}{(1 + x^2)^2(1 + x)^3}.
\end{aligned} \tag{48}$$

Let the numerator polynomial be $P(x) = x^5 + 5x^3 + x^2 + 2x - 1$. We analyze the roots of $P(x)$ for $x \in (0, +\infty)$:

1. **Existence:** $P(0) = -1 < 0$ and $P(1) = 8 > 0$. By the Intermediate Value Theorem, there exists at least one root $x^* \in (0, 1)$.

2. **Uniqueness:** The derivative $P'(x) = 5x^4 + 15x^2 + 2x + 2$ is strictly positive for all $x > 0$. Thus, $P(x)$ is strictly monotonically increasing on positive reals, implying the root $x^*$ is unique.

Since the mapping $r \leftrightarrow x$ is a bijection, the unique solution $x^*$ corresponds to a unique state ratio $r^*$. Thus, $f_{\text{ANO}}(r)$ changes its convexity exactly once. $\square$

### H.4. Proof of Bounded Gradients

*Proof.* We examine the asymptotic behavior of the gradient derived in Eq. (35). Recall $z = \frac{r-1-\epsilon}{\epsilon}$.

**Case 1:** $r \to +\infty$. This implies $z \to +\infty$. Consequently, $2^{-z} \to 0$ and $2^{-2z} \to 0$. Substituting into Eq. (35):

$$\lim_{r \to +\infty} f'_{\text{ANO}}(r) = \frac{45}{32} \left[ \frac{0}{1+0} - \frac{0}{(1+0)^2} \right] = 0. \tag{49}$$

**Case 2:** $r \to -\infty$. This implies $z \to -\infty$, so $2^{-z} \to +\infty$. We analyze the limit of the terms in Eq. (35):

$$\lim_{z \to -\infty} \frac{2^{-2z}}{1 + 2^{-2z}} = \lim_{y \to \infty} \frac{y^2}{1 + y^2} = 1, \tag{50}$$

$$\lim_{z \to -\infty} \frac{2^{-z}}{(1 + 2^{-z})^2} = \lim_{y \to \infty} \frac{y}{1 + 2y + y^2} = 0. \tag{51}$$

Thus:

$$\lim_{r \to -\infty} f'_{\text{ANO}}(r) = \frac{45}{32} \left[ 2(1) - 4(0) \right] = \frac{45}{16}. \tag{52}$$

Since $f'_{\text{ANO}}(r)$ is continuous on $\mathbb{R}$ and has finite limits at $\pm\infty$, the gradient is globally bounded. $\square$

## I. Proof of Proposition 4.1

**Proposition I.1** (Necessity of Convexity Change). *Let $f(r)$ be a continuous function defined on $\mathbb{R}$. Suppose $f$ satisfies:*

1. ***Bounded Maximization (Principle 2):*** *The set of maximizers is bounded above by $r^*$, and for $r > r^*$, $f(r)$ is strictly decreasing.*

2. ***Asymptotic Stability (Principle 3 & A):*** *The altitude of (sub)gradient decays to $0$ as $r \to +\infty$.*

*Then, $f$ cannot be globally concave on the tail interval $(r^*, +\infty)$. It must exhibit **at least one** change in convexity (inflection point) in this region.*

*Proof.* We proceed by contradiction. Assume that $f(r)$ is globally concave on the interval $(r^*, +\infty)$.

Since $f$ is strictly decreasing on $(r^*, +\infty)$, we can choose two arbitrary points $r_1, r_2$ such that $r^* < r_1 < r_2$. Let $g_2 \in \partial f(r_2)$ be any supergradient of $f$ at $r_2$. By the definition of concavity, we have:

$$f(r_1) \leq f(r_2) + g_2(r_1 - r_2). \tag{53}$$

Rearranging the terms, we obtain:

$$g_2(r_1 - r_2) \geq f(r_1) - f(r_2). \tag{54}$$

Since $r_1 < r_2$, we have $r_1 - r_2 < 0$. Dividing both sides by $(r_1 - r_2)$ reverses the inequality:

$$g_2 \leq \frac{f(r_1) - f(r_2)}{r_1 - r_2}. \tag{55}$$

Let $C = \frac{f(r_1) - f(r_2)}{r_2 - r_1}$. Since $f$ is strictly decreasing, $f(r_1) > f(r_2)$, implying $C > 0$. The inequality becomes:

$$g_2 \leq -C. \tag{56}$$

This inequality holds for *any* $g_2 \in \partial f(r_2)$.

Now, consider an arbitrary $r > r_2$. Let $g \in \partial f(r)$ be any supergradient at $r$. Again, by the definition of concavity applied between $r$ and $r_2$:

$$f(r_2) \leq f(r) + g(r_2 - r) \implies g(r_2 - r) \geq f(r_2) - f(r). \tag{57}$$

Dividing by $(r_2 - r) < 0$:

$$g \leq \frac{f(r_2) - f(r)}{r_2 - r}. \tag{58}$$

From Eq. (56), we know that the secant slope is non-increasing for concave functions. Thus:

$$g \leq \frac{f(r_2) - f(r)}{r_2 - r} \leq \frac{f(r_1) - f(r_2)}{r_2 - r_1} = -C. \tag{59}$$

Consequently, for all $r > r_2$, every supergradient $g \in \partial f(r)$ satisfies $g \leq -C$. Taking the absolute value (since $g$ is negative), we have:

$$|g| \geq C > 0. \tag{60}$$

This implies that $\liminf_{r \to +\infty} |\partial f(r)| \geq C > 0$, which directly contradicts Principle 3 (Asymptotic Stability) requiring $\lim_{r \to +\infty} \partial f(r) = 0$.

Therefore, the assumption must be false. $f$ cannot be globally concave on $(r^*, +\infty)$ and must exhibit at least one change in convexity. $\square$

