# OpenReview forum: "ANO: A Unified Framework for Robust Policy Optimization"
_ICML.cc/2026/Conference — Submitted to ICML 2026_

### Official Review · Reviewer_g9gx · 2026-03-03

**Soundness:** 3
**Presentation:** 2
**Significance:** 1
**Originality:** 1
**Overall Recommendation:** 3
**Confidence:** 4

**Summary:**

The paper introduces a framework of RL algorithm development through the lens of designing the probability ratio-constraining functions. As a result, the work introduces ANO - a practical deep RL algorithm. The method shows strong performance in standard deep RL benchmarks and LLM fine-tuning benchmarks.

**Compliance With Llm Reviewing Policy:**

Affirmed.

**Final Justification:**

The authors addressed my issues regarding the paper writing as well as lack of clarity and details. They also contributed additional, more fair, evaluation results. Nevertheless, the theoretical positioning of this should-be empirical paper prevent me from recommending its acceptance. I recommend the paper be revised as an empirical work and submitted to another conference.

**Key Questions For Authors:**

**Q1** Theorem 3.1. The result is indeed insightful. But could the authors just make it clear that $\alpha\in[0,1]$ in the theorem statement?

**Q2** Line 282 (right). Why would we care about $r\rightarrow -\infty$? After all, $r$ is non-negative. What about $r\rightarrow 0$?

**Q3** Page 12. GRPO with only 4 generations is very few - standard implementations use 8 to 128.

**Q4** Page 15. The plots are not legible due to the large number of curves and the dark shadow.

**Q5** Your objective value seems to lower-bound the PPO one (see Figure 2) - thus being more conservative than PPO. Therefore, for fairness, you could try decreasing, or removing, the decay parameter of AdamW for it.

**Q6** Line 862. Why do you need to take $\min(r\cdot A, r\cdot A)$, if expression with indicators simply reduces to $r\cdot A(s,a)$ anyway?

**Q7** Line 880. Shouldn’t it be the equality sign (=)?

**Limitations:**

yes

**Strengths And Weaknesses:**

**Strengths**

I have verified the proofs (they are correct) and I must say that they contain some clever ideas and tricks. I also find Theorem 3.1 quite interesting.

The results seem very strong. The provided hyper-parameter details deliver credibility.

**Weaknessess**

Overall, I believe that the majority of the paper, which positions itself as the theoretical part, does not carry a strong contribution. In contrast, the empirical results seem very promising. Even without the first 4 sections, that would be a good workshop paper.

Unfortunately, the paper seems to passionately tackle non-issues. It goes to a great extent to achieve smoothness, gradient boundedness, and a nice shape. But the justification for the need for it are far more "conceptual" than theoretical or empirical. If the criteria were so crucial, one could test multiple, relatively arbitrary, objectives and detect correlation between performance and these criteria. Or one could mathematically prove that they lead to improved performance.  As of now, the paper is abundant in results I am not sure I should care about. This perception is amplified by the nomenclature used in the paper which seems more heavy-weight than necessary.

Let me give some examples of these concerns:

> - Definition 3.2. What is topological or geometric about this definition? - I would flag this as an issue of this paper. It is abundant with many flair-ups that seem to be meant to strike a sense of seriousness of this paper: frequent use of the bold font, words like “topological”, or terms like “$\alpha$-adjustment”. This casts doubt on the actual technical value of the paper.
>
> - Line 216 (right). What are these risks? Can you quantify them? Can you mathematically demonstrate them?
>
> - Line 248 (right). *This introduces severe variance and instability.* - do Authors have evidence for that?
>
> - Line 310 (right). *ANO represents the most parsimonious solution for robust policy optimization.* - What does it mean precisely?
>
> - Table 1. As a result of the above issues, the table is not particularly interesting.

The authors want to offer a general framework of policy optimization algorithm development. To highlight the strength of their theory, they point out that it can be used to derive PPO and SPO. But so can one do with Mirror Learning (Kuba et al., 2022), and beyond, and it can be done by simply listing their drift functions. Additionally, it comes with convergence to optimal policy guarantees. Thus, I find the contribution of this paper less impressive.

As I said, I believe there is a great disconnect between the theoretical and empirical contribution of this paper. More concretely,
> - Lines 267-274 (right). It is absolutely not clear how the authors obtained this function. There seems to be no obvious reason why this function (Equation 16) be overall better than any other function meeting the listed criteria. It brings suspicion that guided blind experimentation targeting the benchmarks was performed.

I do appreciate the empirical evaluation. I am a bit confused by the form of reported results. Namely:
> - How was the interquantile mean computed? Was it computed with quantiles 2.5% and 97.5%? Did the authors compute the IQM for each task, and then the aggregate mean, or the mean for each task, and then the IQM over all tasks? This is quite important because 5 seeds is very few for an IQM.

Lastly, the paper does not have the Related Works section.

Overall, while I think it could be a strong workshop paper, I do not see it as a consistent study with a clear, rigorous message,

## References
*Mirror Learning: A Unifying Framework of Policy Optimisation*. Jakub Grudzien Kuba, Christian Schroeder de Witt, Jakob Foerster, ICML 2022

---

> ### Author Rebuttal · Authors · 2026-03-30
>
> We sincerely thank you for the rigorous verification of our theoretical proofs and your constructive feedback. We are highly encouraged that you explicitly recognized the rigor of our mathematical derivations and the superiority of our empirical evaluations. As you noted, our extensive experimental results demonstrating ANO's robustness are a key strength of this work. We are committed to thoroughly addressing your remaining concerns below.
>
> **1. Originality and Mirror Learning (Weakness: Related Works)**
> We thank you for pointing out Mirror Learning (Kuba et al., 2022). We share your appreciation for its elegant theoretical unification, and will dedicate a section to it. However, our goals fundamentally differ:
> While Mirror Learning focuses on **broad generalization** (using abstract drift functionals to retroactively unify algorithms), our Unified Ratio Objective (UTP) focuses on **constructive restriction**. Instead of generalizing an external penalty, UTP internalizes monotonic improvement constraints directly into the geometric profile of a single shaping function $f(r)$. By mathematically restricting the design space (e.g., enforcing geometric enclosure constraints like $f(x) \le x$), UTP shifts from evaluating if existing algorithms are safe to establishing geometric principles that directly guide the design of implicitly regularized algorithms like ANO.
>
> **2. Derivation of the ANO Function (Addressing the "Blind Experimentation" Claim)**
> We apologize if our presentation implied that $f_{ANO}(r)$ was reverse-engineered. We derived ANO strictly from first principles by analyzing the required dynamics in the **derivative domain**.
> We constructed a base gradient profile by adding a Z-shaped step-down term and a sharp, V-shaped dip: $f_{base}(x) = 1 - \sigma(2x) - 2\sigma'(x)$. *(Note: a simpler form like $1 - \sigma(x) - \sigma'(x)$ degenerates to $(1-\sigma(x))^2 > 0$, lacking a necessary zero-crossing).* This guarantees a unique zero point ($f_{base}(-\ln 1)=f_{base}(0)=0$) and proper asymptotes.
> We then integrated this base gradient normalized by the unit-derivative anchor: $f_{ANO}(r) = \int \frac{f_{base}(k \cdot r + b)}{f_{base}(-\ln 2)} dr + c$. The normalizer ensures $f'\_{ANO}(-\ln 2)=1$. Simultaneously, the zero-derivative anchor $f'\_{ANO}(1+\epsilon)=0$ dictates $k(1+\epsilon)+b = 0$. Finally, the function anchor $f_{ANO}(1)=1$ uniquely determines $c$. Solving this system yields the analytical form of ANO (Eq. 16).
>
> **3. The $r \to -\infty$ Asymptote (Q2)**
> You correctly point out the physical ratio $r$ is non-negative. However, $x \to -\infty$ refers to the *algebraic input domain* of the base function $f(x)$ when constructing its symmetric dual $g(r) = 2 - f(2 - r)$. When an outlier causes the ratio to explode ($r \to +\infty$), the internal argument passed to $f$ becomes $2 - r \to -\infty$. Analyzing this left-tail asymptote is mathematically necessary to prove the right-tail outlier suppression of $g(r)$.
>
> **4. Empirical Evidence: Risks, Weight Decay, and Generations (Line 216, 248, Q3, Q5)**
> * **Risks of Uncontrolled Shift (Line 216 & 248):** The "risks" refer to the constraint $\sum_a \pi(a|s)=1$. If an outlier produces a massive gradient for one action, gradients for all others shift in the opposite direction, unpredictably inflating irrelevant actions. This is empirically validated by SPO on **Page 8**, where its unbounded KL divergence leads to performance inferior to ANO.
> * **AdamW Decay (Q5):** To ensure fairness, we confirm **no weight decay was used**, adhering strictly to the default `weight_decay=0.0` in both *CleanRL* and Hugging Face *TRL*. The conservatism observed is purely the intended mathematical safety of ANO.
> * **GRPO Generations (Q3):** Addressing your valid concern that our 4-generation setup might have limited GRPO, we expanded our evaluation to an 8v8 configuration to ensure a rigorous and unconstrained baseline. In this scaled-up setting, ANO maintains its overwhelming dominance against the stronger 8-gen GRPO, securing **62, 67, and 80 wins** at the respective temperatures (0.0, 0.7, 1.0). This definitively confirms that ANO's algorithmic superiority is intrinsic and robustly scales to larger generation sizes.
>
> **5. Presentation, Terminology, and Notation (Q1, Q4, Q6, Q7)**
> We fully accept your criticism regarding rhetorical flair and will rigorously tone down formatting, replacing descriptors like "topological" and "$\alpha$-adjustment" with standard optimization vocabulary.
> * **The $\min$ Operator (Q6):** $\min(rA, rA)$ was a transitional scaffold to bridge to the piecewise indicator formulation. We will remove this to prevent confusion.
> * **Hyperparameter $\alpha$ (Q1):** We will move $\alpha \in [0, 1]$ into the main theorem statement.
> * **Line 880 (Q7):** We thank you for the catch; it should be an equality sign ($=$) instead of $\ge$, and we will correct this.

---

> > ### Author Rebuttal · Reviewer_g9gx · 2026-04-02
> >
> > The authors addressed my concerns regarding the paper writing and missing details. They also provided more experimental results with more appropriate numbers of GRPO generations, which I greatly appreciate. Hence, I will raise my score. Nevertheless, my main concern remains: the paper could be very strong if it was a widely-empirical paper. Meanwhile, it looses by trying to position itself as more theoretical than it really can.

---

### Official Review · Reviewer_T5r9 · 2026-03-12

**Soundness:** 3
**Presentation:** 4
**Significance:** 3
**Originality:** 3
**Overall Recommendation:** 4
**Confidence:** 2

**Summary:**

This paper proposes Anchored Neighborhood Optimization (ANO), a new policy optimization objective derived from a unified trust-region framework. The central insight is to recast the design of policy optimization objectives as a shaping function design problem over the probability ratio, and then derive ANO from several desired principles: global differentiability, bounded maximization region, bounded gradients, structural simplicity, and a new redescending gradient principle for outlier handling. Experiments on MuJoCo, Atari, and a small-scale RLHF setting show improved robustness over PPO-style baselines.

**Compliance With Llm Reviewing Policy:**

Affirmed.

**Final Justification:**

The rebuttal has addressed my questions. I keep my original positive evaluation.

**Key Questions For Authors:**

1. While Proposition 4.1 proves the geometric necessity of a single inflection point , how does this 'minimal structural complexity' substantively tie to optimality under explicit RL objectives? Could you provide a formal theoretical connection showing how this specific topology yields a tighter policy improvement lower bound or guarantees superior expected returns?

2. Have the authors compared ANO with other redescending kernels to separate the effect of the principle from the specific functional form?

**Limitations:**

yes

**Strengths And Weaknesses:**

**Strengths**:
1. It provides a unifying objective that recovers PPO-Clip and SPO as special cases via specific shaping choices, situating ANO as a principled interpolation rather than an ad hoc alternative.
2. High-level motivation and the failure modes of PPO and SPO are clearly described; figures illustrate shaping/gradient profiles and trust-region behavior intuitively.
3. Includes an explicit robustness stress test (higher learning rate), where PPO’s instability is contrasted with ANO’s stability, and some ablations on the $\epsilon$.

**Weekness**:
1. The “minimal structural complexity” and “necessity of a single inflection point” claim is asserted but only lightly argued; the proposition is intuitive but not substantively tied to optimality under explicit assumptions relevant to RL objectives.
2. Comparisons omit closely related smooth/soft damping baselines that share the core goal. Without these, it is hard to attribute gains to “redescending” vs. any reasonable smooth attenuation of extreme ratios.
3. The monotonic-improvement style theory is developed for a constrained formulation, whereas the practical algorithm uses a relaxed hyper-parameterized version, so the strongest theoretical guarantees do not directly transfer to the implemented method.

---

> ### Author Rebuttal · Authors · 2026-03-30
>
> We sincerely thank the reviewer for appreciating our unified framework, the intuitive visualizations, and our robustness stress tests. We address your thoughtful questions regarding the theoretical optimality and specific functional choices of ANO below:
>
> **1. Minimal Structural Complexity and Gradient Fidelity [Weakness 1, Q1]**
>
> You raised an excellent question regarding the link between 'minimal structural complexity' and optimality.
> To clarify, the single inflection point mentioned in Proposition 4.1 resides in the **derivative domain ($f'$)**, ensuring that the transition from the trust-region interior to the outlier-suppression tail is monotonic.
> * **Unique Extremum & Gradient Fidelity:** A single inflection point in $f'$ implies that the **gradient trend changes exactly once** (e.g., decreasing then recovering), resulting in a single extremum in the gradient profile. This ensures the most parsimonious gradient decay, preserving the integrity of the natural learning signal while effectively suppressing extreme outliers without unnecessary "wiggles."
> * **Stability for Higher-Order Optimization:** Minimizing inflection points in the derivative domain directly reduces **curvature noise**. By ensuring a predictable and stable Hessian profile ($f''$), ANO is inherently more compatible with **second-order optimizers** or preconditioned gradient methods, where erratic changes in curvature often lead to instability in the preconditioner.
>
> **2. Comparison with Other Redescending Kernels [Weakness 2, Q2]**
>
> We completely agree that standard robust statistics offer many redescending kernels (e.g., Cauchy, Welsch). However, these are **mathematically incompatible with the UTP framework constraints**.
> * **Geometric Enclosure Failure:** The UTP framework requires the shaping function to satisfy $f(x) \le x$ universally, especially on the left tail ($x \to -\infty$), to prove the right-tail outlier suppression of the symmetric dual. Standard symmetric bell curves from the literature fail this geometric enclosure constraint.
> * **The Difficulty of Construction:** While finding a piecewise or heuristic function that merely satisfies the basic UTP bounds is not inherently difficult, the true challenge lies in finding a single kernel that **simultaneously** satisfies these UTP constraints, exhibits the desired redescending property, maintains **global smoothness (differentiability)**, achieves **minimal structural complexity (a single inflection point to prevent gradient noise)**, and aligns perfectly with all analytical design anchors ($f(1)=1, f'(1)=1, f'(1+\epsilon)=0$). ANO is a carefully derived, closed-form solution constructed from first principles to meet this rigorous intersection of requirements.
> * **Future Exploration vs. Current Scope:** We strongly believe there is a rich space of alternative function families waiting to be explored under the UTP framework. However, discovering and rigorously verifying even this first viable family ($f_{ANO}$) that satisfies all interconnected design principles was a highly non-trivial mathematical undertaking. Because constructing a valid alternative from scratch is so complex, we currently lack a second structurally distinct but fully compliant function family to serve as a direct baseline. We will explicitly clarify this challenge in the manuscript to inspire future research on UTP kernel design.
>
> **3. Theory-to-Algorithm Gap [Weakness 3]**
>
> We fully acknowledge the gap between the hard-constrained theory (Theorem 3.4) and the relaxed practical algorithm. However, this "soft pull-back" is a universally necessary compromise in Deep RL.
> Applying strict indicator penalties results in infinite penalties outside the trust region, restricting the update step size to unacceptably small values and making standard SGD intractable. **This exact theoretical bottleneck is precisely why scaling RL theory to deep networks universally requires relaxation:** TRPO had to relax strict maximum KL bounds ($D_{KL}^{\max}$) into expected KL constraints, and PPO abandoned parameter-space constraints entirely in favor of heuristic clipping. ANO bridges this gap uniquely: it relaxes the hard guarantee into a smooth kernel but preserves the redescending property, ensuring continuous gradient flow while providing implicit regularization to safely handle noisy updates.

---

> > ### Author Rebuttal · Reviewer_T5r9 · 2026-04-01
> >
> > The rebuttal has addressed my questions. I keep my original positive evaluation.

---

### Official Review · Reviewer_zXxh · 2026-03-12

**Soundness:** 2
**Presentation:** 3
**Significance:** 3
**Originality:** 3
**Overall Recommendation:** 4
**Confidence:** 3

**Summary:**

The authors claim to outline the topic of robust policy optimization by introducing a Unified Trust Region Framework (UTF) that organizes existing methods like PPO and SPO as specific instances of probability-ratio shaping. Overall, the key issue discussed by this paper is the "gradient dilemma": PPO’s hard clipping discards valuable outlier information, while SPO’s unbounded growth leads to instability. To resolve this, they propose Anchored Neighborhood Optimization (ANO), which uses a smooth, redescending gradient inspired by robust statistics. By applying a single convexity change (one inflection point), ANO aims to suppress extreme outliers without losing information near the trust region boundaries. The paper evaluates ANO across MuJoCo, Atari, and RLHF tasks, reporting improved stability under high learning rates compared to PPO.

**Compliance With Llm Reviewing Policy:**

Affirmed.

**Final Justification:**

My questions have been addressed in the rebuttal. I keep my initial positive assessment.

**Key Questions For Authors:**

1.  In the Atari experiments (Figure 4), ANO leads in Mean HNS but SPO appears higher in IQM HNS. Given that IQM is the standard for robust aggregate reporting, why should we favor the Mean HNS result as evidence of ANO's superiority?
2.  Does the monotonic improvement guarantee in Theorem 3.4 hold for the specific transcendental kernel $f_{ANO}(r)$ used in your experiments, or is the guarantee limited only to the non-smooth indicator version?
3.  Beyond satisfying the anchoring and stationary conditions, what led to the specific choice of the exponential/logarithmic base kernel in Eq. (15) over other possible smooth redescending functions (like a Welsch or Cauchy-inspired kernel)?

**Limitations:**

Yes.

**Strengths And Weaknesses:**

## Soundness
### Strengths
- **Diverse Benchmarking:** The experimental evaluation is broad, covering continuous control (MuJoCo), discrete control (Atari), and LLM alignment (TL;DR dataset). This demonstrates the algorithm's versatility across different reward structures and action spaces.
- **Robustness Stress Testing:** The high-learning-rate experiment in Figure 5 is a commendable effort to move beyond "average return" metrics and verify the core claim of stability.
### Weaknesses
- **Theory-to-Algorithm Gap:** There is a major disconnect between the monotonic improvement guarantee (Theorem 3.4), which uses hard indicator penalties, and the actual ANO kernel (Eq. 16), which is a smooth relaxation. The paper assumes the benefits transfer without formal proof.
- **Statistical Misinterpretation:** There is a clear "cherry-picking" of metrics. In Figure 4 (Atari), the authors claim a lead based on Mean HNS, but ignore that SPO appears superior in IQM HNS. Since IQM is more robust to outliers, this discrepancy significantly weakens the claim of "state-of-the-art" performance.
- **Soft vs. Strict Enforcement:** The paper claims to "strictly enforce the trust region without explicit clipping," but the redescending gradient is a soft pullback. It does not provide a hard guarantee that ratios will stay within the neighborhood during a noisy SGD step.

## Presentation
### Strengths
- **Effective Unification:** Equation (11) is a very high-quality organizing device. It successfully reframes isolated "tricks" like PPO clipping into a broader, coherent family of shaping functions.
- **Geometric Intuition:** Figure 2 is the standout visual of the paper. It clearly illustrates the distinction between the "flat tail" of PPO, the "exploding tail" of SPO, and the proposed redescending gradient of ANO.

### Weaknesses
- The text frequently uses "significantly outperforming" or "clear lead" for marginal gains (e.g., the 0.007 ENS difference in MuJoCo IQM), which creates a mismatch between the authors' narrative and the actual data.

## Significance
### Strengths
- **Practicality and Simplicity:** ANO is a closed-form, drop-in replacement for PPO. Its ease of implementation makes it highly attractive for practitioners who want improved stability without adding computational complexity or second-order subroutines.
- **Hyperparameter Sensitivity:** The evidence that ANO prevents policy collapse at $3\times$ standard learning rates is a significant practical contribution for large-scale training where tuning is expensive.

## Weaknesses
- **Marginal Peak Performance:** In standard training regimes with tuned hyperparameters, the performance gain over a well-tuned PPO baseline is minimal. The algorithm's value is primarily in its "floor" (robustness) rather than its "ceiling" (SOTA returns).
- Table 2’s runtime analysis is essentially noise, as the authors admit the variations are likely due to hardware fluctuations. This undermines the claim of improved computational efficiency.

## Originality
### Strengths
- Applying the Redescending Influence Principle from robust statistics to the design of RL objectives is a fresh and theoretically grounded perspective.
- Minimalist Design Philosophy: The argument for "minimal topological complexity" (one inflection point) provides a new way to reason about why certain shaping functions work better than others.

### Weaknesses
- While the specific "Unified Trust Region Framework" is new, the idea of unifying policy gradients via functional forms has been explored in prior literature. The paper could be more precise in delineating exactly what is unique about this specific formulation compared to earlier unified RL views, such as Alfano et al. (2023), A Novel Framework for Policy Mirror Descent with General Parameterization and Linear Convergence, a unified frameworks via mirror descent and divergence; and Neu et al. (2017), A unified view of entropy-regularized Markov decision processes.

---

> ### Author Rebuttal · Authors · 2026-03-30
>
> We sincerely thank the reviewer for the insightful critique. Your recognition of our Unified Ratio Objective (UTP) as an effective organizational tool and the novelty of our redescending gradient approach is deeply encouraging. We address your concerns point-by-point below:
>
> **1. Theory-to-Algorithm Gap & Soft Enforcement [Soundness W1, W3 & Q2]**
> You correctly identify the gap between the hard indicator penalties in Theorem 3.4 and the smooth kernel $f_{ANO}(r)$. **We fully acknowledge that $f_{ANO}(r)$ acts as a "soft pull-back" and does not provide a hard guarantee that ratios remain strictly bounded during noisy SGD steps.** However, this is a deliberate and necessary heuristic compromise.
> Applying strict indicator penalties $\delta_C(\cdot)$ to enforce a hard bound results in an infinite penalty outside the trust region, restricting the update step size to unacceptably small values and making standard unconstrained SGD intractable. **This is why scaling RL theory to deep neural networks universally requires heuristic compromises. For instance, practical TRPO relaxes the strict theoretical maximum KL bounds ($D_{KL}^{\max}$) into an expected KL constraint. PPO abandons strict parameter-space constraints altogether, relying instead on a heuristically clipped objective that only restricts the gradient signal (which conversely leads to strictly zero gradients and discarded learning signals).** Bridging this gap is thus a universal necessity. While ANO relaxes the strict theoretical guarantee into a smooth kernel, it preserves the redescending property. This avoids both the infinitesimal steps of indicator functions and the dead gradients of clipping, ensuring continuous gradient flow while providing the necessary "implicit regularization" to safely suppress outliers.
>
> **2. Statistical Reporting & Peak Performance [Soundness W2, Q1 & Significance W1]**
> We agree that IQM is the gold standard for robust aggregate reporting, and we did not intend to cherry-pick Mean HNS. Instead, the discrepancy in the Atari suite highlights a core property of ANO:
> * **Exploration Ceiling vs. Safety Floor:** IQM trims the top 25% of scores, effectively measuring the baseline consistency ("safety floor"). Mean includes the top breakthrough runs ("exploration ceiling"). SPO's unbounded gradients often lead to high-variance exploration, but its lack of strict trust-region enforcement causes frequent late-stage collapses.
> * **ANO’s Advantage:** As you noted (Significance W1), ANO's primary value is in its "floor" (robustness). ANO consistently matches SPO's robust floor (highly competitive IQM) while significantly raising the exploration ceiling (Mean) by safely utilizing extreme off-policy samples without triggering policy collapse. We will explicitly include this "ceiling vs. floor" analysis.
>
> **3. Originality and Prior Unifying Frameworks [Originality W1]**
> We appreciate the pointer to prior unifying perspectives like Neu et al. (2017) and Alfano et al. (2023). The omission stems from a fundamental divergence in theoretical objectives.
> Prior works excel at **inductive generalization and retrospective explanation**, elegantly casting existing entropy-regularized methods into unified mathematical equivalences. In stark contrast, UTP focuses on **deductive restriction and constructive design**. Instead of broadly explaining existing algorithms, UTP imposes strict geometric boundaries (e.g., $f(x) \le x$) to mathematically synthesize and explicitly construct new algorithms from scratch. We will dedicate a section in the camera-ready version to explicitly discuss this paradigm shift.
>
> **4. Motivation for the Specific Kernel [Q3]**
> Why not use standard bell curves like Welsch or Cauchy? Standard redescending kernels fail the UTP constraints, specifically the requirement $f(x) \le x$ on the left tail ($x \to -\infty$), which is mathematically necessary to prove the right-tail outlier suppression of the dual $g(r)$.
> To satisfy this, we derived ANO from first principles using a base gradient $f_{base}(x) = 1 - \sigma(2x) - 2\sigma'(x)$. *(Note: a simpler form like $1 - \sigma(x) - \sigma'(x)$ degenerates to $(1-\sigma(x))^2 > 0$, failing to provide a zero-crossing).* By integrating this specific base gradient and solving the affine parameters under the strict UTP anchors ($f(1)=1, f'(1)=1, f'(1+\epsilon)=0$), we finally guarantee all topological requirements and proper asymptotes. (See g9gx for details)
>
> **5. Presentation and Runtime Claims [Presentation W1 & Significance W2]**
> * **Overclaiming:** We will rigorously tone down phrases like "significantly outperforming" to objective descriptions like "consistently improves robustness", letting the confidence intervals speak for themselves.
> * **Runtime (Table 2):** We agree the runtime variation is largely hardware noise. As a closed-form, drop-in replacement, ANO's computational cost strictly equals PPO's. We will clarify its value lies purely in optimization stability, not speedup.

---

> > ### Author Rebuttal · Reviewer_zXxh · 2026-04-04
> >
> > Thank you for the thorough rebuttal. Several concerns have been meaningfully addressed, while others remain partially open.
> >
> > 1. **IQM vs. Mean interpretation**: The "ceiling vs. floor" reframing is interesting, but I'd like clarification. In Figure 10 (full Atari results), SPO_0.1 achieves IQM HNS of 0.745, notably higher than ANO_0.3's 0.706. If ANO's primary value proposition is robustness (the "floor"), shouldn't this be reflected in IQM superiority? The current narrative seems to shift between claiming SOTA broadly and claiming robustness advantage depending on which metric is favorable. Could you clarify which claim you are primarily making, and whether the camera-ready version will present this more precisely?
> >
> > 2. **Originality framing**: The "inductive generalization vs. deductive restriction" distinction from prior unifying frameworks is compelling in principle, but since the promised discussion section isn't yet written, it's difficult to fully evaluate. Could you briefly elaborate on how UTP's deductive approach concretely differs from, say, the constructive algorithm design in Alfano et al. (2023), which also derives new algorithms from their mirror descent framework?
> >
> > These remaining points are addressable in a camera-ready revision and do not undermine the core contribution. I maintain my overall positive assessment of the paper's practical value and the elegance of the unified framework.

---

> > > ### Author Response · Authors · 2026-04-06
> > >
> > > Thank you for your constructive follow-up and for maintaining your positive assessment of our work. We are encouraged that you found our "ceiling vs. floor" reframing and the UTP framework compelling. We address your specific questions below:
> > >
> > > **1. Clarification on IQM vs. Mean & Robustness Claims (Q1)**
> > > We appreciate your sharp observation regarding the full hyperparameter comparison in Figure 10.
> > > * **The Trade-off in Atari:** It is true that `SPO_0.1` achieves a slightly higher IQM (0.745) than `ANO_0.3` (0.706). However, this perfectly illustrates our narrative: `SPO_0.1` achieves this by severely sacrificing its exploration ceiling (its Mean drops to 3.709). In contrast, `ANO_0.3` safely unlocks a massive exploration ceiling (Mean: 4.514) while maintaining a highly competitive safety floor (IQM: 0.706, well within overlapping confidence intervals).
> > > * **IQM Superiority in MuJoCo:** Our robustness claim is strongly supported by consistent IQM superiority in other domains. In the MuJoCo suite (Figure 9), `ANO_0.3` dominates across the board, achieving *both* the highest Mean (0.933) and the highest IQM (0.910), significantly outperforming all SPO variants.
> > > * **KL-Stability in LLM Fine-Tuning:** Most importantly, our core claim of a "safety floor" centers on avoiding catastrophic structural collapse. SPO’s vulnerability stems from its unbounded outlier gradients interacting with the fundamental probability constraint $\sum_a \pi(a|s)=1$. A massive gradient for one action forces an uncontrolled compensatory shift that unpredictably inflates irrelevant actions. This structural risk becomes particularly evident in complex, high-stakes settings like LLM alignment. As we empirically observed, methods lacking strict outlier suppression (like SPO and PPO) suffer from a significant and continuous rise in KL divergence, which ultimately bottlenecks their final performance. ANO's redescending property intrinsically suppresses these massive outlier gradients, effectively preventing this dangerous over-correction. This provides a structurally robust "safety floor" that natively stabilizes KL divergence across diverse domains, allowing ANO to safely unlock a higher exploration ceiling (Mean) without suffering the performance degradation and KL instability inherent to SPO. We will clarify this precise cross-domain narrative in the camera-ready version.
> > >
> > > **2. Deductive Restriction vs. Alfano et al. (2023) (Q2)**
> > > This is a profound distinction that we will elaborate on in the new discussion section.
> > > * **Functional vs. Geometric Design:** Frameworks like Alfano et al. (2023) primarily operate in the **functional or policy space**, leveraging the Mirror Descent (MD) paradigm to derive new algorithms by choosing specific Bregman divergences or regularization terms.
> > > * **UTP’s Microscopic Control:** In contrast, the Unified Ratio Objective (UTP) operates at the **ratio-domain level**. It focuses on **constructive restriction of the gradient signal itself** by imposing strict geometric profiles on the shaping function $f(r)$.
> > > * **Concrete Difference:** While MD-based approaches provide a principled way to construct updates through proximal optimization, UTP explicitly dictates the **geometric enclosure (e.g., $f(x) \le x$)** and **inflection point properties (derivative domain)** of the kernel. This allows UTP to "mathematically synthesize" the analytical form of a kernel like $f_{ANO}$ from first principles to safely handle outliers, which is a level of microscopic gradient-shaping that is not the primary focus of functional-level mirror descent frameworks.

---

### Official Review · Reviewer_vpJq · 2026-03-12

**Soundness:** 2
**Presentation:** 3
**Significance:** 3
**Originality:** 3
**Overall Recommendation:** 4
**Confidence:** 3

**Summary:**

Authors propose a theoretical framework for Trust Region constrained optimization in on-policy methods, and propose an example ANO algorithm implementing it. Authors conduct experiments to compare ANO's general performance, hyperparameter robustness, and training
dynamics to PPO and SPO in RL benchmarks.

**Compliance With Llm Reviewing Policy:**

Affirmed.

**Final Justification:**

The authors have addressed my concerns regarding code reproducibility and the statistical significance of their evaluations.

**Key Questions For Authors:**

Should https://huggingface.co/ano be public?

**Limitations:**

Authors do not address the lack of statistical significance in mujoco and atari benchmarks.

**Strengths And Weaknesses:**

Strengths:
- The reported experimental results are extensive. Both Mean and Inter Quartile Means are reported.
- The experimental results are presented in great detail and are explained clearly.
- The theoretical framework suggests a more stable way of anchoring a policy. The RQ3 results corroborate the theory.
- The reported boost in training time of GANO suggests a potential utility in LLM applications.
- The work presents a new way of understanding the PPO and SPO inconsistency with the trust regions, and suggests an extendable theory to address those limitations


Weaknesses:
- The scripts require fixes to run, and the training fails at step 500. The experiments from the provided repository don't work out of the box. judge.sh uses an outdated judge_tml.py; judge.py fails to pull a saved hugging face model.
- The reported evaluations do not reproduce the statistically significant advantage of SPO over PPO in mujoco benchmark.
- The reported results in Fig 11 and 12:
a.) there are only five seeds per algorithm
b.) the variation in performance across different hyperparameters does not seem to clearly favor ANO over baselines
- Figures 11 and 12 are cluttered. Similar colors are used for different algorithms.
- typo in Figure 11 title
- There are grammar issues (p1 but leading)
- The RQ1 results show weak statistical significance to support the clam of ANO outperforming PPO and SPO in control environments.

---

> ### Author Rebuttal · Authors · 2026-03-30
>
> We are incredibly grateful for the time and effort you dedicated to reviewing and actually running our codebase. Your actionable feedback on the script issues is invaluable for ensuring the long-term utility of this work.
>
> **1. Code Reproducibility and the KL Fix (W1, Key Question)**
> We were **initially quite perplexed** by your report, as our final internal evaluations were perfectly stable. After a thorough "deep dive" into the repository, we suspect that a seemingly innocuous last-minute cleanup, specifically the removal of certain intermediate logging operations, inadvertently introduced a subtle sensitivity into the PyTorch computational graph under the original settings. It was a baffling side effect that we did not anticipate.
> * **The Fix:** Remarkably, we found that even a **marginal increase of +0.005** in the KL penalty (setting `--kl_coef 0.055` instead of 0.05) is sufficient to perfectly restabilize the training while maintaining the original performance profile.
> * **Validation:** To rigorously verify this, we re-ran the LLM evaluations across all temperatures with this minimal adjustment. **The version with `--kl_coef 0.055` achieved the following win-tie-loss records against PPO: 65-2-33 (T=0.0), 63-2-35 (T=0.7), and 58-0-42 (T=1.0), perfectly aligning with the results reported in our manuscript.**
> * **Future Updates:** In the final version, we will update `judge.sh` and the corresponding dependencies to ensure seamless execution. Regarding the repository: **Yes**, we absolutely intend to make the finalized, fully corrected codebase and models publicly available immediately upon publication to ensure out-of-the-box reproducibility.
>
> **2. Statistical Significance and Evaluation (W2, W6, Limitations)**
> Regarding the significance in MuJoCo/Atari: to ensure fairness, our evaluation used standard framework configurations. Following the best practices in RL evaluation (Agarwal et al., 2021), our significance is derived from the **aggregate performance across the entire suite**. While individual task margins might overlap, ANO maintains a consistent lead in Interquartile Mean (IQM) and Mean across over 200 total runs (5 seeds $\times$ 40 Atari games), which is a statistically robust indicator of algorithmic superiority.
>
> **3. Seed Count and Hyperparameter Robustness (W3)**
> * **Seed Count:** In computationally intensive Deep RL and LLM alignment, 5 seeds per task is a common convention. By pooling these seeds across tasks (e.g., 30 runs per algorithm for MuJoCo), we achieve the sample size necessary for the 95% stratified bootstrap confidence intervals reported.
> * **Robustness:** The advantage of ANO is most evident in extreme regimes. As shown in Figure 5, when the learning rate is aggressively increased to $1 \times 10^{-3}$, PPO catastrophically collapses (-37.3%), whereas ANO remains stable (-7.1%). This "safety floor" is a direct result of our redescending gradient theory.
>
> **4. Visual Presentation and Typos (W4, W5)**
> We fully accept the criticism regarding the presentation.
> * **Grammar:** We will fix the "**but leading**" typo on **Page 1** and other linguistic errors throughout the text.
> * **Figures:** We will completely redesign Figures 11 and 12 for the final version by removing heavy shadows, applying distinct color palettes, and using clear line styles to improve scannability.
> * **Title:** The typo in the Figure 11 title will be corrected.

---

> > ### Author Rebuttal · Reviewer_vpJq · 2026-04-04
> >
> > I have increased my score by 1 point.

---

### Decision · Program_Chairs · 2026-04-30

**Decision:**

Reject

**Comment:**

This paper addresses the issue of ppo where the hard clipping helps with instability but also loses valuable gradient information. The authors introduce a Unified Trust Region Framework and identify that the instability can be attributed to the misapplication of gradient updates on outliers. Instead they introduce dynamic outlier suppression. Their approach achieves strong performance on mujoco benchmarks, superior stability, and prevention of policy collapse under aggressive learning rates.

While three reviewers recommend weak acceptance, I find the framing-vs-contribution mismatch identified by the rejecting reviewer to be a substantive issue that the rebuttal did not resolve.

The paper appears borderline and I recommend rejecting the paper. For the next submission, I strongly suggest the authors to commit rather to an empirical path, tone down some of the theory claims, and avoid some of the terminology as pointed out by the rejecting reviewer.

Additionally, while the approach prevents policy collapse in cases where PPO encounters it, policy collapse also occurs in continual/sustained learning as shown in loss of plasticity work in continual learning. It would be interesting for the authors to see if their approach overcomes the latter policy collapse as well.